# An injury-associated lobular microniche is associated with the classical tumor cell phenotype in pancreatic cancer

Sara Söderqvist[1], Annika Viljamaa [1], Natalie Geyer[1], Anna-Lena Keller[1], Kseniya Ruksha[1], Carina Strell [2,3], Neda Hekmati[3], Alexandra Niculae[1], Jennie Engstrand [1], Ernesto Sparrelid[1], Caroline Salmén[1], Tânia D. F. Costa [4], Miao Zhao [3], Staffan Strömblad [4], Argyro Zacharouli[5], Poya Ghorbani[1], Sara Harrizi[1], Yousra Hamidi[1], Olga Khorosjutina [6], Stefina Milanova[6], Bernhard Schmierer [6], Béla Bozóky[5], Carlos Fernández Moro [1,5,7,9] & Marco Gerling [1,8,9] ✉

Pancreatic cancer is an aggressive disease with a dense fibrotic stroma and is often accompanied by chronic inflammation. Peritumoral inflammation is typically viewed as a reaction to nearby tumor growth. Here, we report that the inflamed pancreatic lobules are frequently invaded by tumor cells, forming a distinct, non-fibrotic tumor niche. Using a semi-supervised machine learning approach for annotations of clinical samples and multiplex protein profiling, we show that tumor cells at the invasion front are closely associated with acinar cells undergoing damage-induced changes, and with activated fibroblasts expressing markers of injury. The invaded lobules are linked to classical tumor phenotypes, in contrast to fibrotic areas where tumor cells display a more basal profile, highlighting microenvironment-dependent tumor subtype differences. In female mice, lobular invasion similarly aligns with the classical tumor phenotype. Together, our data reveal that pancreatic tumors colonize injured lobules, creating a unique niche that shapes tumor characteristics and contributes to disease biology.

Pancreatic ductal adenocarcinoma (PDAC) is the main type of pancreatic cancer and has one of the lowest survival rates among solid tumors[1]. Histologically, PDAC is dominated by extensive desmoplastic stroma surrounding nests of tumor cells[2]. This desmoplastic microenvironment is thought to result from stromal cell proliferation and extracellular matrix deposition[2], mainly driven by tumor cell-derived signals[3–7]. An extensive body of research has revealed heterogeneous PDAC stromal populations[2], such as inflammatory cancer-associated fibroblasts (iCAFs) and myofibroblastic CAFs (myCAFs), which have partly opposite roles in PDAC progression[8,9].

On the tumor cell level, two main transcriptomic PDAC subtypes have been identified, "basal" (or "basal-like") and "classical"[10–13].

[1]Department of Clinical Science, Intervention and Technology – CLINTEC, Karolinska Institutet, Huddinge, Sweden. [2]Centre for Cancer Biomarkers - CCBIO, Department of Clinical Medicine, University of Bergen, Bergen, Norway. [3]Department of Immunology, Genetics and Pathology, Uppsala University, Uppsala, Sweden. [4]Department of Medicine, Huddinge - MedH, Karolinska Institutet, Huddinge, Sweden. [5]Department of Clinical Pathology and Cancer Diagnostics, Karolinska University Hospital, Stockholm, Sweden. [6]CRISPR Functional Genomics, SciLifeLab and Department of Medical Biochemistry and Biophysics, Karolinska Institutet, Solna, Sweden. [7]Department of Laboratory Medicine, Division of Pathology, Karolinska Institutet, Stockholm, Sweden. [8]Theme Cancer, Karolinska University Hospital, Solna, Sweden. [9]These authors contributed equally: Carlos Fernández Moro, Marco Gerling. ✉e-mail: marco.gerling@ki.se

Predominantly basal tumors have particularly poor outcomes and are largely resistant to chemotherapy[12]. Within individual tumors, basal and classical cells and their hybrid forms coexist in a spectrum of cell states[14,15]. Spatially resolved data have begun to connect tumor cell features to distinct stromal niches[15], suggesting that microenvironmental factors are essential drivers of tumor cell phenotypes. These observations imply significant microenvironmental control of the tumor cell phenotype and reveal tumor microenvironment interactions as important determinants of tumor cell behavior.

Unlike PDAC, the initial stages of its precursor lesion, pancreatic intraepithelial neoplasia (PanIN), lack a desmoplastic stroma, which develops in advanced PanIN stages[16]. In mice, oncogenic transformation of pancreatic acinar or duct cells (i.e., their progression into precancerous or cancerous states) precedes desmoplasia formation by days to weeks[17,18]. Together, these data establish a chronological sequence where the oncogenic transformation of epithelial cells foregoes stromal activation. In contrast, the spatial sequence of events during the continuous invasion of cancer cells in established PDAC is largely unknown. Unlike other tumors, such as liver metastases, where a clear invasion front can morphologically be defined[19], PDAC growth is more diffuse and accompanied by regenerative and inflammatory changes in the peritumoral pancreas, which may impair accurate tumor cell identification[20]. Consequently, areas at the interface between invading PDAC cells and non-malignant pancreatic tissue are incompletely defined.

Here, using multimodal, cross-species analyses of resected human tumors and murine models, we show that PDAC colonizes and invades the pancreatic lobules, where an area of acinar injury and non-desmoplastic stromal activation defines the invasive tumor edge. Machine learning (ML)-augmented image analyses of lobular invasion reveal that human PDAC cells colocalize with a specific subset of non-malignant epithelial cells undergoing acinar-to-ductal metaplasia (ADM), a regenerative program linked to injury and stromal activation[21]. Quantification of tumor cell markers connects the classical phenotype to lobular regions and a specific stromal cell subset, while tumor cells in the archetypical desmoplastic regions adopt a more basal phenotype. In mice, orthotopic injection of near-clonal tumor cells reveals a switch from the basal to the classical phenotype in areas of lobular invasion, indicating a role for the microenvironmental regulation of the PDAC phenotype in pancreatic lobules. Together, our results reveal the colonization of injured pancreatic lobules as a route of tumor invasion and connect the injury-induced lobular microenvironment to distinct tumor cell phenotypes. These results have implications for our understanding of PDAC biology and stromal evolution, and might inform clinicopathological diagnostics.

## Results

### Colonization of pancreatic lobules defines a distinct route of PDAC invasion

PDAC cell nests are characteristically embedded in an archetypical desmoplastic stroma. However, in pathological routine diagnostics, we frequently observed that tumor cells also invaded the pancreatic lobules, the major anatomic units of the pancreas (Fig. 1a). In the lobules, tumor cells were seen next to the main lobular cell types, such as acinar cells, the endocrine islets of Langerhans, intralobular ducts, and slender connective tissue (Fig. 1b)[22].

To quantify the frequency of lobular invasion, we reviewed all available histological slides from a cohort of $n = 108$ patients operated for PDAC at Karolinska University Hospital (Supplementary Fig. 1; patient identification strategy described in "Methods"). Because the peritumoral pancreas frequently features chronic pancreatitis and reactive epithelial changes, which impair the accurate identification of tumor cells in this complex tissue context[23], we employed

immunohistochemistry (IHC) for tumor protein p53 (p53) and SMAD family member 4 (SMAD4) to identify tumor cells. In *TP53*-mutated cells, p53 accumulates; upon *SMAD4* mutation, SMAD4 protein is lost (Fig. 1c), which allowed us to confirm the presence of tumor cells in the different tissue compartments in most tumors. We detected lobular invasion in $n = 103$ tumors (95.4%, Fig. 1d). PDAC cells in the lobules were seen at varying extents and accompanied by different degrees of lobular atrophy, as assessed by a pancreas pathologist (C.F.M., Supplementary Fig. 2a–c). For a subset of $n = 5$ tumors selected to represent different degrees of lobular invasion, we quantified the fraction of all tumor cells in the lobules (tumor$^{in\_lobules}$) and in the stroma (tumor$^{in\_stroma}$). To this end, we used whole-slide images (WSIs) of hematoxylin and eosin (H&E) stains and IHC for p53, and manually annotated for tumor cells in each compartment. We defined tumor$^{in\_stroma}$ as areas devoid of any lobular remnants, such as endocrine islets of Langerhans, which typically remain discernible even in severe or end-stage lobular atrophy and serve as unequivocal indicators of lobular origin (Supplementary Fig. 2d, e). Tumor$^{in\_lobules}$ was defined as tumor cells within pancreatic lobules, localized next to pancreatic parenchymal cells, such as intralobular ducts, acinar (exocrine) cells, acinar cells in the process of ADM as assessed morphologically, and/or endocrine islets of Langerhans. The proportion of lobular invasion ranged from 4% to 32% (mean: 14.8%, standard deviation [SD]: 10.8%, Fig. 1e). As expected, tumor$^{in\_stroma}$, the histopathological archetype of PDAC, was the predominant tumor location; however, tumor$^{in\_lobules}$ emerged as a prevalent compartment of PDAC invasion (schematic in Fig. 1f).

### The classical PDAC cell subtype dominates in the pancreatic lobules

The canonical PDAC subtypes, classical and basal, are subject to microenvironmental control, which contributes to intratumoral subtype heterogeneity[24]. In the lobular microenvironment, tumor cells are not surrounded by a desmoplastic stroma. Instead, they are adjacent to non-malignant stromal and epithelial cells, reminiscent of the replacement-like pattern of tumor growth described in other organs[25]. Since the lobular microniche is composed of various cell types not present in the desmoplastic stroma, we hypothesized that tumor cells in the lobules are likely exposed to different external signaling cues than tumor cells in the stroma, which may influence their cellular phenotypes[15,24].

Independent studies have shown that a limited number of markers can approximate tumor phenotype identity on the single-cell level[26,27]. Hence, to study compartment-related tumor cell phenotypes, we used multiplex IHC with a panel of six markers, two classical (caudal type homeobox 2 [CDX2], mucin 5AC oligomeric mucus/gel forming [MUC5AC]), and four basal markers (Keratin [KRT]5, high mobility group AT-hook 2 [HMGA2], Carbohydrate antigen [CA]125/MUC16, and KRT17; Fig. 2a) in a subcohort of $n = 31$ patients, for which extensive IHC was available and lobular invasion was present at different degrees (clinical characteristics in Table 1, flow chart of sample analysis in Supplementary Fig. 1). We annotated tumor cell regions on IHC-immunolabelled WSIs and quantified positive tumor cells for each marker and compartment (Fig. 2b, c and Supplementary Fig. 3a)[28]. We found a strong positive correlation between the classical markers, MUC5AC and CDX2, and between the basal markers, KRT5, HMGA2, CA125, and KRT17, respectively (Supplementary Fig. 3b), supporting the validity of this panel to call PDAC subtypes.

Comparing classical and basal markers between tumor$^{in\_lobules}$ and tumor$^{in\_stroma}$ revealed significantly lower fractions of cells positive for the basal markers, CA125, KRT17, and HMGA2, in the lobules (Fig. 2d and Supplementary Figs. 4–7; adjusted $p$-values: 3.795e-3, 4.206e-05, and 6.120e-03, respectively). In contrast, the fractions of tumor cells positive for the classical markers, MUC5AC and CDX2, were higher in tumor$^{in\_lobules}$ compared to tumor$^{in\_stroma}$ (Fig. 2d and Supplementary

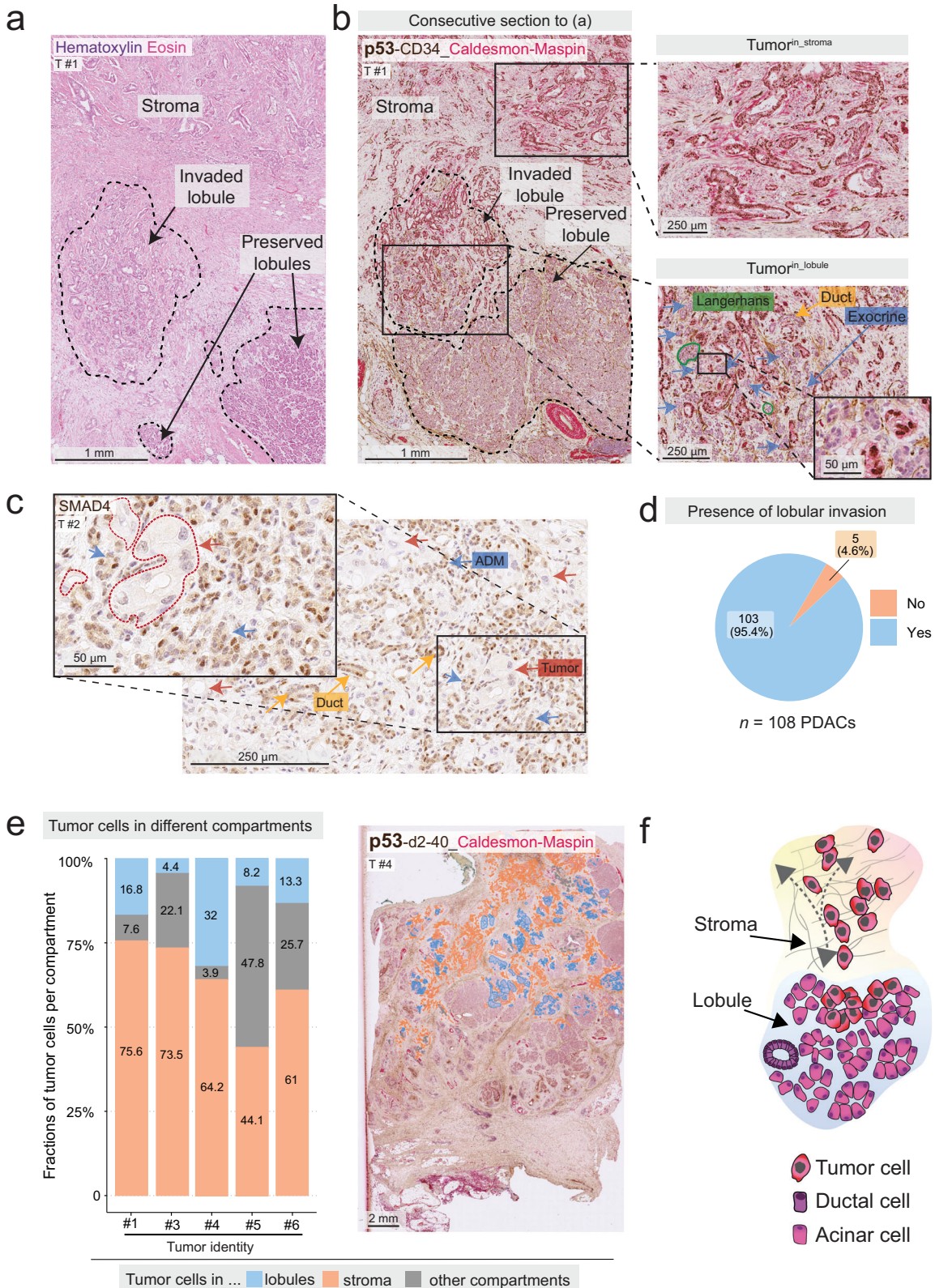

Figs. 8 and 9; adjusted *p*-values: 1.071e-4 and 2.620e-3, respectively). In line with this, unsupervised hierarchical clustering of IHC quantifications separated the regions of interest (ROIs) derived from the lobular compartment from those derived from the stromal compartment (Fig. 2e). Compartment-associated heterogeneity, defined as at least one marker significantly differently expressed, was present in *n* = 24 PDACs (77%); of

these *n* = 10 tumors differed significantly in one marker, *n* = 10 in two to three markers and *n* = 4 in four or more (Supplementary Figs. 4–9).

To assess the overall degree of heterogeneity in all regions irrespective of lobular or stromal location, we used the Simpson diversity index, an established heterogeneity metric[29,30]. Here, the Simpson index describes the probability of retrieving two random ROIs with

**Fig. 1 | Pancreatic ductal adenocarcinoma colonizes the pancreatic lobules.**
**a** Hematoxylin and eosin (H&E) stain of a pancreatic ductal adenocarcinoma (PDAC) in desmoplastic stroma (upper part, "stroma"), and pancreatic lobules (lower part, dashed lines). **b** Immunohistochemistry (IHC) labelling of tumor protein p53 (p53, nuclear, brown) of a consecutive section to (**a**). Dashed lines: pancreatic lobules. Upper insert: p53⁺ tumor cells in the desmoplastic stroma. Lower insert: In the pancreatic lobules, PDAC is adjacent to the indicated lobular epithelial structures. Green lines: endocrine islets of Langerhans, blue arrows: exocrine cells, yellow arrows: ducts. **c** Annotated tumor nests (red dotted line) with loss of SMAD family member 4 (SMAD4, nuclear and cytoplasmic, brown), growing next to acinar-to-ductal metaplasia (ADM) structures. Non-malignant acinar/ADM cells are distinguished from tumor cells by SMAD4 positivity and morphology. **d** All available IHC and H&E sections from n = 114 PDACs operated at Karolinska University

Hospital were assessed for the presence of lobular invasion. Six tumors were excluded due to insufficient sampling. Out of n = 108 tumors, n = 103 tumors had lobular invasion. **e** Tumor content in whole-slide images from n = 5 tumors per tissue compartment: In stroma, lobules, and other locations. The largest tumor fraction is located in the stroma (range 44.1%–75.6%). Lobular invasion is present in all tumors (range 4.4%–32%). "Other": Pancreatic ducts, common bile duct, adipose tissue, nerves, vessels (range 3.9%–47.8%). Right panel: example of the overlayed annotations of each area class. **f** Depicted schematically, tumor cells localize to the stroma-transformed compartment, or the pancreatic lobules. **a–c** Representative of n = 31 tumors from n = 31 patients. **b, c, e** Additional markers used, but not analyzed here, are cluster of differentiation (CD)34 (brown), D2-40 (brown), caldesmon (red), and maspin (red), with hematoxylin counterstain. Source data are provided in the Source Data file.

---

different fractions of positive tumor cells of each respective subtype marker, allowing us to assess overall heterogeneity per marker. We found a weak correlation of the Simpson index with compartment-associated differential expression between tumor[in_lobules] and tumor[in_stroma] (Supplementary Fig. 10a, b). This suggests that while tumor cell location in lobules *vs.* stroma contributes to PDAC subtype identity as one of several factors, it is not the exclusive driver. These findings are in line with previous studies that have identified factors such as *KRAS* gene dosage, the presence of specific mutations in other genes, and stromal interactions as tumor phenotype modulators[31–33].

Taken together, these data revealed that pancreatic lobules represent a distinctive tumor compartment enriched for classical-type tumor cells, whereas the basal tumor phenotype dominates in the stroma.

## Machine learning–based tissue profiling reveals tumor cell localization within a chronic pancreatitis-like lobular microenvironment

Tumor[in_stroma] represents the archetypical histological pattern of PDAC, where tumor cells are embedded in stromal cells and a rich extracellular matrix (ECM)[22]. Tumor cells in lobules meet different cellular partners, such as epithelial cells and a distinctive type of non-desmoplastic stroma. To quantify cellular interactions in tumor[in_lobules] *vs.* tumor[in_stroma] in more detail, we trained a convolutional neural network to recognize the primary tissue types on IHC-WSIs using nerve growth factor receptor (NGFR) and cluster of differentiation (CD)146 to differentiate stromal and epithelial structures (Fig. 3a). Based on NGFR/CD146 expression and cellular morphology, the model was trained to recognize acinar cells, exocrine cells undergoing ADM (i.e., cells in an acinar tissue context that had changed morphological appearance toward a more ductal phenotype), ductal cells, immune infiltrates, fibroblasts and ECM (combined to a "fibrosis" class), vessels, nerves, endocrine islets of Langerhans, and tumor cells (Fig. 3a and Supplementary Fig. 11a). Training and the ML-based annotations were further manually curated by two pathologists (C.F.M., B.B.), resulting in an efficient and precise approach to identify the complex tissue structures. The resulting tissue type annotations (n = 9326 areas from n = 41 ROIs in n = 5 tumors) identified tumor cells and all other classes, allowing us to quantify the non-tumor tissue types adjacent to PDAC cells in situ, defined as proximity within a 20 µm radius of the tumor cells' nuclei (Fig. 3b). After benchmarking against manual-only annotations (Supplementary Fig. 11b), we quantitatively compared tumor cell adjacency to all tissue types in tumor[in_lobules] (n = 15 ROIs) *vs.* tumor[in_stroma] (n = 26 ROIs). We found that PDAC cells in both compartments were most frequently adjacent to "fibrosis" (Fig. 3c). However, tumor[in_lobules] was less frequently associated with the "fibrosis" class (97% in tumor[in_stroma] vs. 80.5% in tumor[in_lobules], Fig. 3c). Apart from the fibrotic context, tumor cells within lobules were predominantly adjacent to ADM cells, whereas in desmoplastic

regions, they were more frequently located near nerves and vessels, consistent with the perineural and (peri-)vascular invasion typical of PDAC (Fig. 3d)[34].

To follow ADM emergence in progressive lobular atrophy and tissue remodeling, we co-labelled n = 8 sections with PDAC from n = 8 individual tumors with the ductal marker, KRT19, and the acinar marker, α-amylase (AMY). We found KRT19 and AMY double positive cells, representing early/intermediate-stage ADM, to be most abundant in lobules with minimal to moderate atrophy (Fig. 3e). In contrast, lobules in states of severe or end-stage atrophy, as per pathologist assessment, were significantly enriched in AMY⁻/KRT19⁺ ductally committed cells. Accordingly, acinar cell abundance (AMY⁺/KRT19⁻) decreased with increasing atrophy (Fig. 3e and Supplementary Fig. 12).

In summary, tumor invasion into the lobular compartment is accompanied by ADM, and related to the degree of lobular atrophy. Acinar cells displaying morphological signs of ADM are the primary epithelial cell type in lobules invaded by cancer cells, defining the leading edge of PDAC tumor invasion into the lobules as a zone of chronic pancreatitis-like lobular injury.

## A distinct population of injury-associated NGFR⁺ fibroblasts localizes to the leading edge of lobular tumor invasion.
Despite prevalent contacts of tumor[in_lobules] with the exocrine epithelium, tumor-fibrosis contacts were frequent in the lobules (Fig. 3c). Using the ML-annotated data to generate spatial maps of tumor and non-tumor cells in their spatial context, we found that in lobular tumor invasion, juxtapositions of PDAC cells and fibrosis were enriched at the periphery of the lobules, while tumor cell/ADM and tumor cell/acinar contacts peaked at the leading edge of the tumor-exocrine cell border (Fig. 4a and Supplementary Fig. 13a, b). Assuming that the periphery of the tumor bulk in the lobules (i.e., the tumor-exocrine border) is the first site of tumor-pancreas contact, the data suggested that stromal transformation of lobules increases with tumor invasion, prompting a more detailed investigation into the lobular stroma.

NGFR is upregulated in benign fibrotic conditions and has been associated with a favorable prognosis in PDAC[35–38]; its stromal expression has been linked to pancreatitis[39]. During routine pathology assessment of clinical samples, we consistently observed strong stromal NGFR expression in lobular regions exhibiting features of chronic pancreatitis and morphological signs of tissue injury (Fig. 4b). Analysis of publicly available single-cell RNA sequencing (scRNA-seq) data from mice with acute experimental pancreatitis (Fig. 4c)[40] revealed an enrichment of *NGFR*⁺ stromal cells during acute pancreatic injury (Fig. 4d), a condition associated with the emergence of ADM[40]. *NGFR* expression overlapped with the expression of the iCAF marker, platelet-derived growth factor receptor alpha (*PDGFRα*), but not with the antigen-presenting (ap) CAF marker, *CD74*, and the myCAF marker, alpha-smooth muscle actin (ASMA, encoded by *ACTA2*, Fig. 4e). These data suggest that NGFR⁺

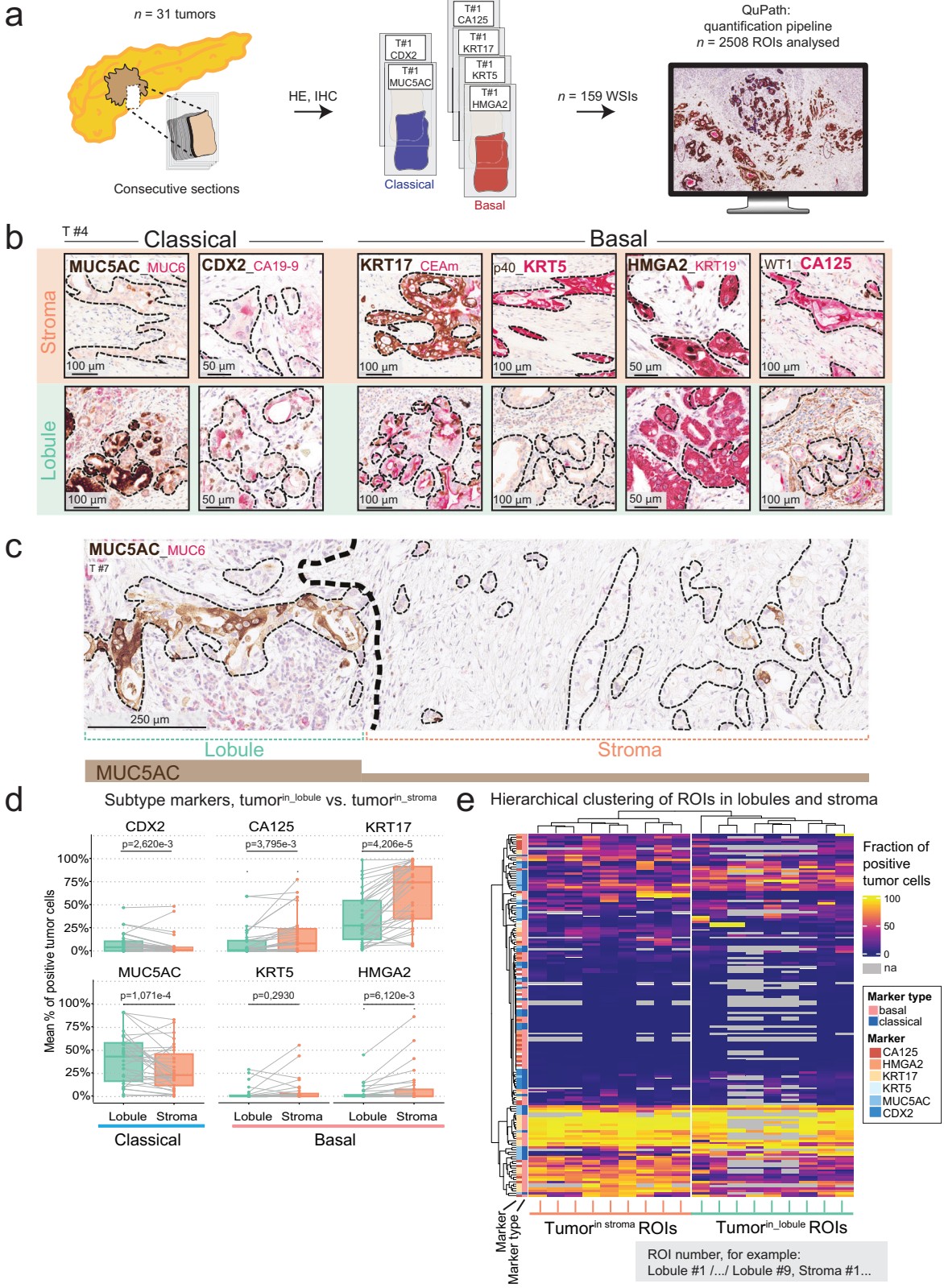

cells are associated with lobular injury both in murine models of pancreatitis and in human PDAC.

To quantify NGFR protein expression in relation to tumor invasion in human PDAC, we utilized the ML−augmented annotations to measure NGFR surrounding individual tumor cells ($n = 5103$; Supplementary Fig. 13a), taking into account their distance from the tumor-exocrine invasive edge. Within invaded lobules, NGFR abundance

decreased as the distance between tumor cells and the intralobular tumor-exocrine interface increased (Fig. 4f). To further quantify NGFR expression in relation to progressing tumor cell invasion, we first measured NGFR abundance in morphologically unaffected lobules without inflammation and without overt acinar changes to establish an expression baseline in near-normal pancreatic lobules (Supplementary Fig. 13a). Next, we compared NGFR expression between lobules

**Fig. 2 | Pancreatic tissue compartments drive PDAC subtype identity.**
**a** Schematic of sampling to analyze pancreatic ductal adenocarcinoma (PDAC) subtypes in stromal vs. lobular compartments. Consecutive sections from $n = 31$ tumors were stained with immunohistochemistry (IHC) ($n = 2$ "classical" makers and $n = 4$ "basal" markers). WSIs: whole-slide images. ROIs: regions of interest.
**b** Representative IHC of subtype-related expression patterns of tumor cells in stroma (upper panel) or lobule (bottom panel). From left to right: classical markers, mucin 5AC oligomeric mucus/gel forming (MUC5AC, brown, cytoplasmic) and caudal type homeobox 2 (CDX2, brown, nuclear) and basal markers, Keratin (KRT) 17 (brown, cytoplasmic), KRT5 (red, cytoplasmic), high mobility group AT-hook 2 (HMGA2, brown, nuclear), and carbohydrate antigen (CA)125 (red, cytoplasmic).
**c** Representative IHC of MUC5AC, expressed in lobular invasion (left) and mostly absent in stromal invasion (right). **d** Two-tailed paired Wilcoxon rank sum test with Benjamini–Hochberg (BH) correction for multiple testing of protein expression in stromal vs. lobular compartments. Dots represent the mean fraction of positive

tumor cells per compartment and tumor. Grey lines show paired values for a given tumor. Box-and-Whisker plots show the median (line), the interquartile range (IQR; box), minimum and maximum values within 1.5 times IQR from the first and third quartile (whiskers), and individual data points. BH-corrected $p$-values are given in the Figure. **e** Unsupervised hierarchical clustering of the same data as in (**d**): Fractions of positive tumor cells for each marker, i.e., each IHC image (rows), by individual ROIs in stromal or lobular compartment (columns). Each cell of the heatmap represents the fraction of positive tumor cells for a given tumor for one of the six quantified markers (e.g., T#1, KRT17, T#1, MUC5AC), in a given ROI. A total of 2508 data points were analyzed; 354 values were not available (na), displayed in grey. **b**–**e** Data from $n = 31$ tumors. Thin black dashed lines: tumor. Additional markers included from clinical routine, but not analyzed, are MUC6 (red), CA19-9 (red), carcinoembryonic antigen monoclonal (CEAm; red), p40 (brown), KRT19 (red), and WT1 (brown) with hematoxylin counterstain. Source data are provided in the Source Data file.

## Table 1 | Clinicopathological characteristics of the patients whose tumors were analyzed with multiplex immunohistochemistry

| | Category | Count |
|---|---|---|
| **Age** | Median, years (min-max) | 70 (45–84) |
| **Sex** | Male | 14 |
| | Female | 17 |
| **Tumor stage** | T1 | 0 |
| | T2 | 16 |
| | T3 | 14 |
| | T4 | 1 |
| **Nodal stage** | N0 | 3 |
| | N1 | 12 |
| | N2 | 16 |
| **Metastatic stage** | M0 | 22 |
| | M1 | 7 |
| | n/a | 2 |
| **Lymphatic vessel infiltration** | pL0 | 0 |
| | pL1 | 31 |
| **Venous infiltration** | pV0 | 4 |
| | pV1 | 26 |
| | n/a | 1 |
| **Perineural infiltration** | pPn0 | 2 |
| | pPn1 | 29 |
| **Resection margin** | R0 | 10 |
| | R1 | 21 |
| **Tumor location** | Caput | 25 |
| | Collum | 1 |
| | Cauda | 5 |
| **Neoadjuvant chemotherapy** | Yes | 5 |
| | No | 26 |
| **Relapse** | Yes | 25 |
| | No | 6 |
| **Time-to-relapse** | Median, months (min-max) | 14 (2–46) |

Tumor, nodal, and metastatic stage according to TNM-8 classification of the pathology report. Resection margin status was determined based on the presence of tumor cells <1 mm from any resection margin. Number of stains quantified for individual tumors are given in the Source Data. *n/a* not available.

exhibiting peritumoral chronic pancreatitis without tumor invasion and those with tumor invasion, which were consistently associated with tissue injury (Fig. 4g and Supplementary Fig. 13c). Stromal NGFR expression was highest in injured lobules exhibiting mild to moderate atrophy, both with and without detectable tumor invasion (Fig. 4g), but declined sharply in the presence of tumor invasion accompanied by advanced atrophy. Together, these findings identify NGFR⁺ inflammation-associated stroma as a feature of lobular injury, contributing to the tumor microenvironment at the lobular tumor-pancreas interface.

**NGFR⁺ lobular fibroblasts overlap only partially with known CAF subtypes in human and murine pancreatic tumors.** Building on established CAF subtype markers in the desmoplastic stroma, ASMA for myCAFs, PDGFRα for iCAFs, and CD74 for apCAFs[9], we further characterized NGFR⁺ stromal cells in situ. To this end, we used multiplex immunofluorescence (m-IF) for ASMA, PDGFRα, CD74, the pan-stroma marker, Vimentin, and the tumor marker, p53 ($n = 8$ tumors, all of which were *TP53* mutated, to identify tumor cells; Fig. 5a and Supplementary Fig. 14a). Three stromal layers were annotated in 15 µm increments, approximating the diameter of a single stromal cell, measured orthogonally from the center of each tumor cell annotation (Supplementary Fig. 14a). Consistent with previous reports[8], ASMA⁺ cells were enriched in the innermost stromal layer adjacent to tumor cell nests, both in stromal and lobular compartments. In contrast, PDGFRα⁺ cells were not enriched in any layer, while CD74⁺ cells were enriched in the innermost layer of tumor^in_stroma regions (Fig. 5b and Supplementary Fig. 14b)[8]. Notably, tumor^in_stroma was consistently deprived of NGFR (Fig. 5c and Supplementary Fig. 14c). However, in lobules with tumor invasion, NGFR⁺ stromal cells co-expressed the iCAF marker, PDGFRα, to a high extent (94%), while 22% of NGFR⁺ cells were positive for ASMA, and 34% for CD74 (Fig. 5d and Supplementary Fig. 15a). This finding is consistent with the co-expression of *NGFR* and *PDGFRα* observed in pancreatitis (Fig. 4d, e). In contrast, PDGFRα⁺ cells surrounding tumor^in_stroma were mostly negative for NGFR (Supplementary Fig. 15b, c). Taken together, these data further support the interpretation that NGFR⁺ stromal cells in the lobular compartment of PDAC are associated with inflammation and tissue injury.

To validate the presence of NGFR⁺ stromal cells in PDAC and further explore their phenotypes, we analyzed public scRNA seq data from a murine orthotopic PDAC injection model[41] and two human PDAC studies[42,43]. All three datasets contained *NGFR*⁺ stromal cells, and most of these cells co-expressed one or more of the CAF markers *ACTA2*, encoding for ASMA, *CD74*, and *PDGFRα*. Among these, *PDGFRα* showed the strongest overlap with *NGFR* expression in all datasets (Fig. 5e). Gene Ontology (GO)-based analyses of *NGFR*⁺ cells revealed the enrichment of immunoregulatory functions in both murine (Fig. 5f) and human (Supplementary Fig. 15d) datasets. Whereas murine NGFR⁺

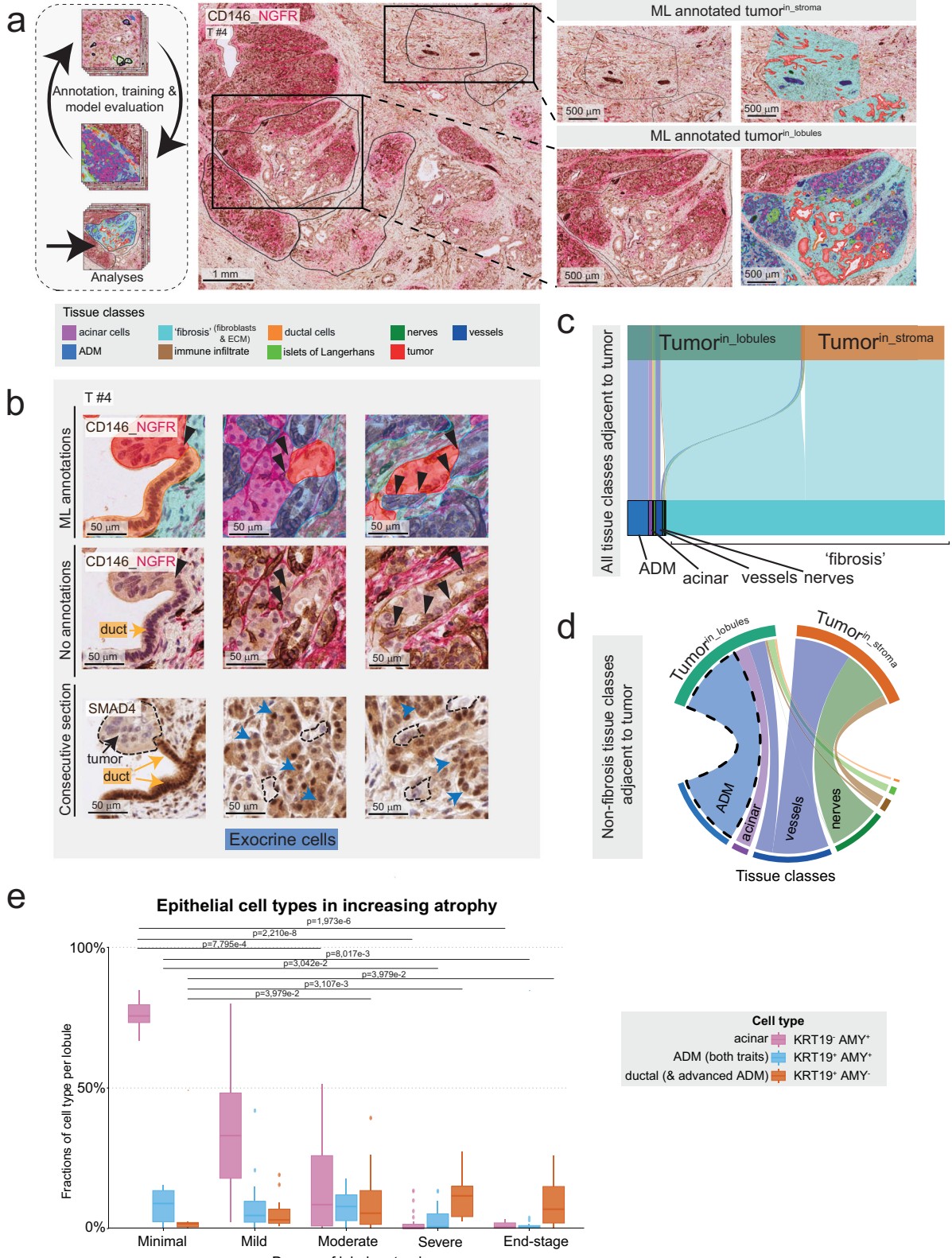

cells predominantly exhibited gene signatures associated with memory T-cell activation and antigen presentation (Fig. 5f), human NGFR+ cells showed enrichment for a broader spectrum of immunoregulatory functions (Supplementary Fig. 15d).

The cells in the analyzed murine dataset were collected at various time points after orthotopic PDAC cell injection, allowing us to assess stromal and tumor cell phenotypes in early (day 10) and later (days 20 and 30) phases of tumor progression (Fig. 5g). The *NGFR* signature, derived from our data (**Source data**) and the iCAF signature[9] were enriched at the early time point (Fig. 5h). At the tumor cell level, classical signatures dominated early after injection, while later time points were enriched in basal and mesenchymal signatures, associated with aggressive disease in murine PDAC[44]. In the human PDAC

**Fig. 3 | PDAC tumor cells in direct proximity to non-malignant epithelial cells at the lobular invasion front. a** A convolutional neural network was trained to recognize the main tissue types on immunohistochemistry (IHC) of cluster of differentiation (CD)146 and nerve growth factor receptor (NGFR, left panel). Detection of multiple classes in stroma (upper panel) and in lobules (lower panel), with overlayed machine learning (ML)-derived classifications (right inserts). ROIs: regions of interest. All results were curated by pathologists before quantitative analyses. **b** Representative examples of proximity between PDAC cells (red) and resident, non-malignant epithelial cells in the lobules, from left to right (upper panel): "duct" (orange), "acinar" (magenta), and acinar-to-ductal metaplasia ("ADM", dodger blue). Cancer cells identified by loss of SMAD family member 4, (SMAD4, brown, nuclear and cytoplasmic) on a consecutive section growing in lobular structures (lower panel). Black arrowheads: tumor/non-tumor contacts in lobular growth; blue arrows: exocrine cells. **c** Distribution of all cell- and tissue classes in proximity to (i.e., within 20 μm of) a tumor cell, stratified by tumor

location and **d** with all cell-and tissue classes other than "fibrosis". **e** Fractions of constituent, non-malignant exocrine cell types according to α-amylase (AMY)$^+$ and keratin (KRT)19$^+$ expression in pancreatic lobules with increasing degree of atrophy, from duplex-immunofluorescence. Data from $n = 8$ tumors, $n = 113$ ROIs in total (minimal atrophy: $n = 7$, mild: $n = 17$, moderate $n = 48$, severe $n = 28$, end-stage $n = 13$). Fractions were calculated from the total cell count in the lobule. Box-and-Whisker plots show the median (line), the interquartile range (IQR, box), and minimum and maximum values within 1.5 times IQR from the first and third quartile (whiskers). Unpaired two-tailed Kruskal−Wallis rank sum test, with Benjamini−Hochberg (BH) corrected post-hoc Dunn's test. BH-corrected $p$-values are given. For readability, $p$-values are only displayed for comparisons with the "minimal" atrophy groups. **a**, **b** Representative of $n = 5$ tumors. Counterstain with hematoxylin. **c**, **d** Data from $n = 5$ tumors; $n = 15$ lobular (tumor$^{in\_lobules}$) and $n = 26$ stromal (tumor$^{in\_stroma}$) ROIs. Source data are provided in the Source Data file.

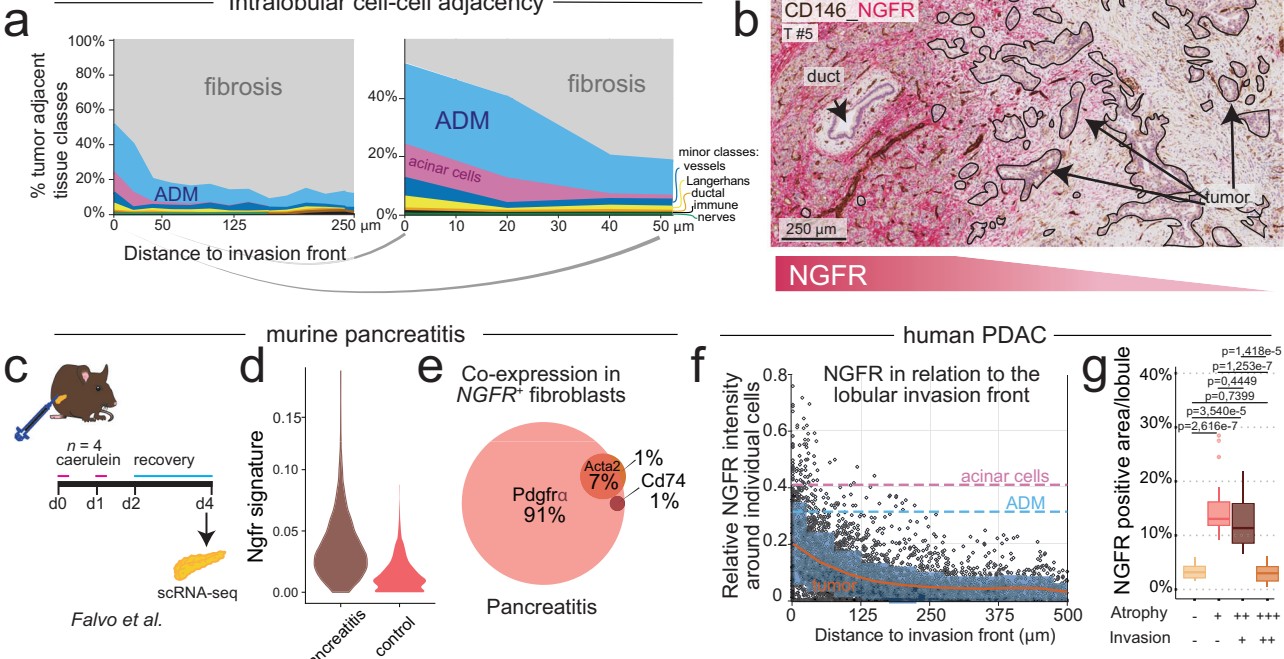

**Fig. 4 | Injury-induced stromal remodeling in lobular invasion. a** Fractions of tumor in proximity (within 20 μm) with other cell- and tissue classes by distance to the lobular invasion front. Right panel: magnification of the first 50 μm from the invasion front. $n = 5$ tumors, $n = 15$ regions of interest (ROIs). **b** Example of immunohistochemistry (IHC) of nerve growth factor receptor (NGFR; red) co-labelled with cluster of differentiation (CD)146 (brown). Representative of $n = 4$ tumors. **c** Schematic of the murine acute pancreatitis model[40]. Mice were injected with either caerulein ($n = 2$) or saline ($n = 2$; control). Three days after the injection, pancreata were retrieved and single-cell RNA sequencing (scRNA-seq) performed. **d** $NGFR$ signature of mice with induced pancreatitis ($n = 2$) or saline-injected control mice ($n = 2$). **e** Proportional Venn diagram of overlapping marker expression with the $NGFR^+$ population from the murine acute pancreatitis model. Data from $n = 124$ $NGFR^+$ cells from $n = 4$ mice. **f** Mean NGFR intensity (IHC quantifications of human

tumors) within a 50 μm radius area expanded from tumor cells ($n = 5103$ cells) in regions of lobular invasion by distance to the lobular invasion front. Orange line: rolling average of NGFR intensity near each individual tumor cell and standard deviation (blue). Pink and cyan dashed lines: averages of NGFR intensity of the expanded area from acinar cells ($n = 5702$) and acinar cells undergoing acinar-to-ductal metaplasia (ADM, $n = 10430$), respectively ($n = 4$ tumors, $n = 12$ ROIs). **g** NGFR$^+$ area of pancreatic lobules with different degrees of atrophy and tumor invasion ($n = 67$ ROIs). Data from $n = 8$ individual tumors. Unpaired two-tailed Kruskal−Wallis rank sum test, with Benjamini−Hochberg (BH)-corrected post-hoc Dunn's test for pairwise multiple comparisons. BH-corrected $p$-values are given. Box-and-Whisker plots show the median (line), the interquartile range (IQR, box), and minimum and maximum values within 1.5 times IQR from the first and third quartile (whiskers). Source data are provided in the Source Data file.

datasets, stromal cells expressing $NGFR$ signatures (**Source data**) were more abundant in smaller, T1 tumors (according to clinicoradiological TNM scoring, which is based on tumor size with T1 ≤ 2 cm, T2 2–4 cm, T3 > 4 cm, Fig. 5i). T1 tumors showed an enrichment of a previously described "normal stroma signature" associated with favorable prognosis[45], while neither classical nor basal tumor cell subtype signatures were significantly enriched in tumors of either T stage (Supplementary Fig. 15e).

Since our data connected NGFR$^+$/$NGFR^+$ cells to inflammatory pathways, we went back to m-IF and analyzed WSIs immunolabelled for macrophage and other immune cell markers (Supplementary Fig. 16a). In particular, macrophage populations have recently been identified as important modulators of the PDAC phenotype and play a role in pancreatic injury[41]. However, we did not observe an enrichment of M2-type macrophages, which are associated with tissue repair (CD68$^+$/CD163$^+$) or CD68$^+$/CD163$^-$ macrophages, approximating the

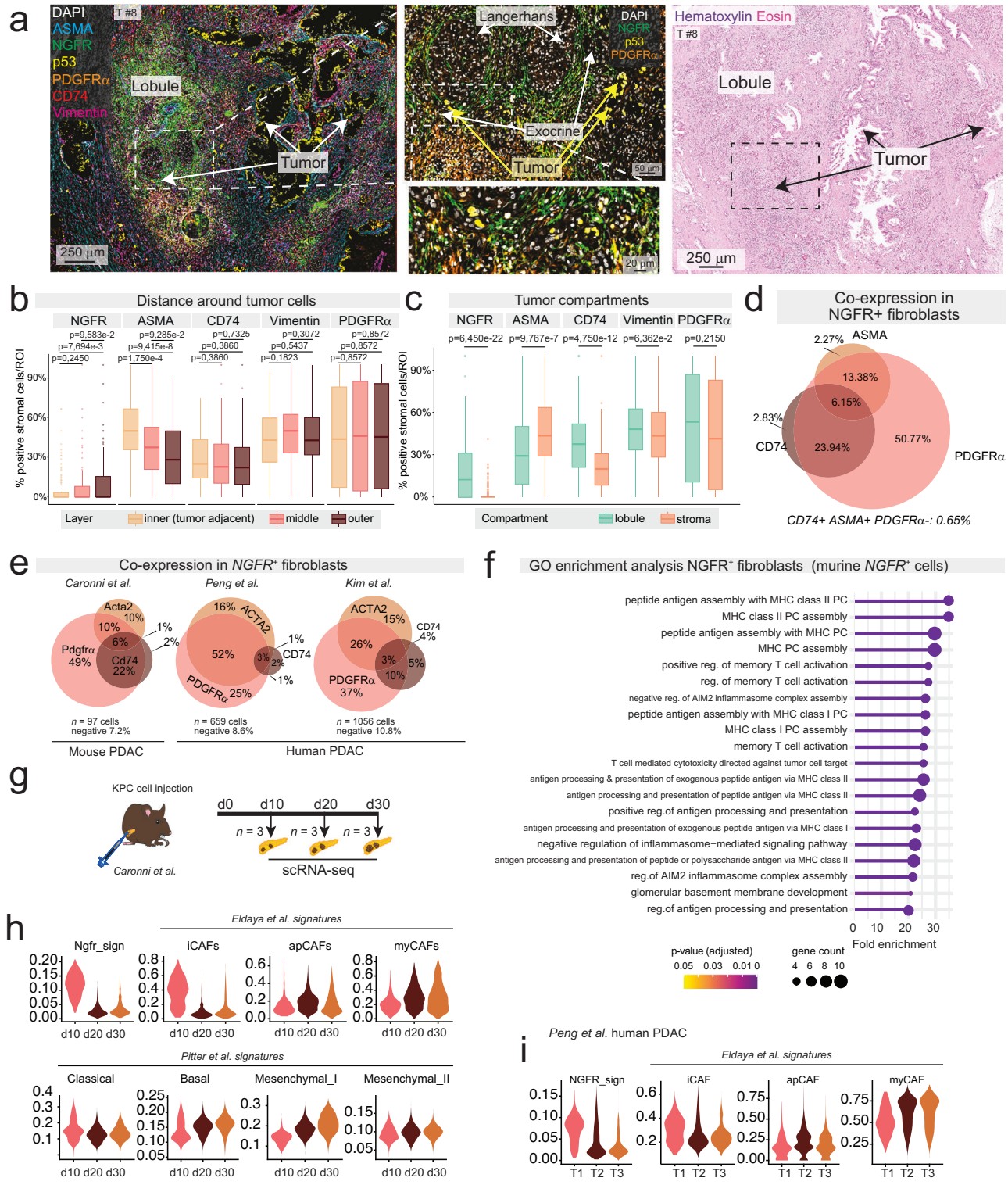

M1 end of the spectrum, in either stromal or lobular locations (Supplementary Fig. 16b).

Together, our m-IF and previously published scRNA-seq data identified a distinct NGFR⁺/PDGFRα⁺ stromal cell population in both human and murine tumors, predominantly localized to inflamed lobules in human PDAC. NGFR⁺ stromal cells were enriched at early time points following tumor injection in mice and were more abundant in smaller human tumors, supporting a model in which stromal NGFR/ *NGFR* expression declines as tumor invasion advances and desmoplastic stroma develops.

**Compartment-driven phenotypic switch in clonal murine pancreatic cancer.** Given that our observations suggested an association between tumor cell phenotypes and their localization to lobular versus stromal compartments, we next employed murine PDAC models to further investigate these spatial–phenotypic interactions in a more controlled setting. The tissue compartment association could primarily reflect a preference for specific tumor clones (defined by tumor intrinsic factors such as mutational background) to colonize lobules. Alternatively, cues from the lobular niche (extrinsic factors) could shape tumor cell phenotypes, consistent with previous reports

**Fig. 5 | NGFR$^+$/PDGFRα$^+$ cells are linked to lobular invasion and an inflammatory microenvironment. a** Representative multiplex-immunofluorescence (m-IF) of stromal markers. Magnifications show tumor$^{in.lobule}$ intermixed with parenchymal structures embedded in NGFR$^+$ stroma. Right panel: corresponding HE region. **b** Expression of NGFR, alpha-smooth muscle actin (ASMA), cluster of differentiation (CD)74, vimentin, and platelet-derived growth factor receptor alpha (PDGFRα). Inner layer: within 15 μm from tumor, middle: 15–30 μm, outer: 30–45 μm. Unpaired two-tailed Kruskal−Wallis rank sum test, with post-hoc Dunn's test. **c** Stromal cell expression within 45 μm to tumor. Unpaired two-tailed Wilcoxon rank sum test. **d**, **e** Proportional Venn diagrams of overlapping marker expression within the NGFR$^+$ population from m-IF (**d**), and with the *NGFR$^+$* populations from murine pancreatic ductal adenocarcinoma (PDAC)[41] (**e**, left; *n* = 977 cells, *n* = 12 mice) and two human PDAC datasets[42,43] (Peng et. al., middle; *n* = 659 cells and *n* = 24 patients, and Kim et al., right; *n* = 1056 cells and *n* = 17 patients). **f** Gene Ontology (GO,

biological processes) enrichment analysis of significantly upregulated genes in *NGFR$^+$* fibroblasts compared to other fibroblasts present in murine PDAC (*n* = 12 mice). Bonferroni−Hochberg (BH)-corrected Fischer's Exact test. **g** Schematic of longitudinal orthotopic injection PDAC model[41], which included *n* = 3 normal pancreata. Cells for injection originated from *Kras$^{G12D/+}$;Trp$^{R172H/+}$;Pdx1$^{cre/+}$*(KPC) mice. scRNA-seq: single-cell RNA sequencing. **h** Signatures overlayed to the longitudinal orthotopic injection model of murine PDAC. Data from *n* = 9 mice. **i** *NGFR* and CAF signatures in human PDAC across tumor size; T1 < 2 cm, T2 2–4 cm, T3 > 4 cm. Data from *n* = 24 patients (Peng et al.). **b**, **c** BH-correction applied. BH-corrected *p*-values are given. Box-and-Whisker plots show the median (line), the interquartile range (IQR, box) and minimum and maximum values within 1.5 times IQR from the first and third quartile (whiskers). *n* = 363 ROIs (222 stromal, 141 lobular; 121 per layer). **a**–**d** Data from *n* = 8 individual tumors. Source data are provided in the Source Data file.

of microenvironmental influences on tumor characteristics[31,41]. Discerning whether clonal selection or niche factors primarily drive the phenotypic divergence could inform strategies to block invasion into either compartment.

To assess whether tumor and stromal phenotypes emerged in parallel and progress gradually, we first analyzed preneoplastic lesions from *Kras$^{G12D/+}$; Pdx1-Cre* (KC) mice, in which PDAC precursor lesions arise through oncogenic *Kras* activation. These were compared to tumors from KPC mice (*LSL-Kras$^{G12D/+}$;LSL-Trp53$^{R172H/+}$;Pdx1-Cre*)[46], which more closely recapitulate human PDAC histopathology and develop a desmoplastic stromal reaction, albeit less extensive than that observed in human disease[47]. As expected, KC mice did not develop invasive tumors but instead presented with pancreatic intraepithelial neoplasia (PanIN) lesions, surrounded by limited stromal activation. We used IF for the basal PDAC phenotype marker, HMGA2, and the classical marker, Galectin-4 (GAL4), to assess tumor phenotypes semiquantitatively (Fig. 6a, b). Sections were co-labelled for the classical marker, GATA binding protein 6 (GATA6), and the mesenchymal marker, VIM (Supplementary Fig. 17a). However, quantitative analysis of GATA6 and VIM expression in mouse tissue was deemed unfeasible due to the widespread expression of GATA6 in both non-malignant epithelial and tumor cells, and the intense stromal labelling of VIM (Supplementary Fig. 17a). Of note, after orthotopic injection, murine scRNA-seq data revealed *GAL4* and *HMGA2* to be expressed at relatively distinct poles corresponding to classical and basal/mesenchymal tumor cell clusters, respectively, while *GATA6* and *VIM* were more broadly expressed (Supplementary Fig. 17b), supporting the usefulness of *GAL4/HMGA2* specifically to call murine PDAC phenotypes in vivo.

We found that HMGA2 protein expression was significantly higher in invasive tumors from KPC mice, whereas it was restricted to a limited number of cells in the precursor lesions of KC mice (Fig. 6c). GAL4 was detected in PanIN lesions but was scarcely expressed in normal ductal cells (Fig. 6a). While no statistically significant difference in GAL4 expression was observed between KC and KPC lesions, KC lesions showed a trend toward higher GAL4 abundance (Fig. 6c). However, high grade, advanced PanINs were significantly higher in HMGA2 and lower in GAL4 expression (Supplementary Fig. 17c, d).

HMGA2 abundance appeared lower in areas of lobular *vs.* stromal tumor location in KPC tumors (Supplementary Fig. 17a, e). However, multifocality and variable latency of tumor formation in the KPC model, where oncogenic hits in most epithelial cells lead to asynchronous, multiclonal tumor development[48], precluded a meaningful analysis of tumor phenotypic markers because the age of each individual lesion cannot be determined.

To further reduce clonality and timing of tumor initiation as a factor for tumor diversity, we next used an orthotopic injection model, where KPCT cells (bearing an additional, inducible tdTomato allele, *B6.Cg-Gt(ROSA)26Sor$^{tm14(CAG-tdTomato)Hze}$/J)*[19] were injected orthotopically

into fully immunocompetent C57BL/6J mice, similarly to the previously analyzed study on murine PDAC (Fig. 5g). First, we lentivirally integrated a library of barcodes into KPCT cells at low multiplicity of infection (Fig. 6d). We injected 10e$^5$ barcoded cells into the pancreata of *n* = 4 mice, collected the established tumors, and counted the barcodes by amplicon sequencing of the lentivirally integrated cassette. In three out of four mice, two clones comprised >90% of the tumor cells (Fig. 6e). In two of those mice, tumors were monoclonal, such that at least 99% of the tumor cells expressed the same individual barcode, originating from one single-cell (Fig. 6e). Five clones comprised >90% of the tumor cells in one case. Hence, orthotopic PDAC cell injection was associated with massive clonal selection, in line with recent data from another injection-based tumor model[49].

H&E stains of the orthotopic model (Fig. 6f) revealed a stroma-rich tumor core, although less dense than in the autochthonous model, possibly owing to the limited time allowed for its development (three weeks compared to approximately three months). Both invasion into the lobular compartment and the presence of ADM adjacent to tumor cells were evident morphologically (Fig. 6f), and further confirmed using the endogenous tdTomato to distinguish tumor cells from host tissue (Supplementary Fig. 18). Next, we quantified IF for the basal tumor marker, HMGA2, and the classical marker, GAL4, and used the acinar marker, AMY, to confirm regions of lobular invasion (Fig. 6g, h, Supplementary Fig. 19a). Consistent with our findings in human PDAC, the basal marker HMGA2 was expressed at lower levels in tumor cells located within lobules compared to those in stromal regions, while GAL4 expression was higher in lobular areas. This resulted in a significant enrichment of classical-like tumor cells in the lobular compartment and basal-like tumor cells in stroma-rich regions (Fig. 6i and Supplementary Fig. 19b).

These findings support a model in which signals from the lobular microniche, comprised of acinar cells, ADM cells, and NGFR$^+$/PDGFRα$^+$ stroma, promote the classical-like phenotype in adjacent PDAC cells. Primary acinar cells can be isolated and cultured in 2D in vitro, upon which they experience cellular stress and undergo ADM over several days[50,51]. This culture system, therefore, has similarities to the acinar cell behavior observed in human and murine PDAC[50,51]. To investigate whether proximity to stressed, isolated acinar cells is sufficient to induce a shift toward a classical and away from a basal tumor cell phenotype, we isolated acinar cells from normal mouse pancreas. These cells partially retained amylase expression in culture through the culture period (Supplementary Fig. 20a). Co-culturing them with KPCT cells established KPCT−acinar/ADM contacts in vitro (Fig. 6j). To assess phenotypic shifts, we quantified IF intensities of HMGA2 (basal marker) and GATA6 (classical) in KPCT cells, identified by their tdTomato expression (Fig. 6j and Supplementary Fig. 20b). HMGA2 expression was significantly higher in

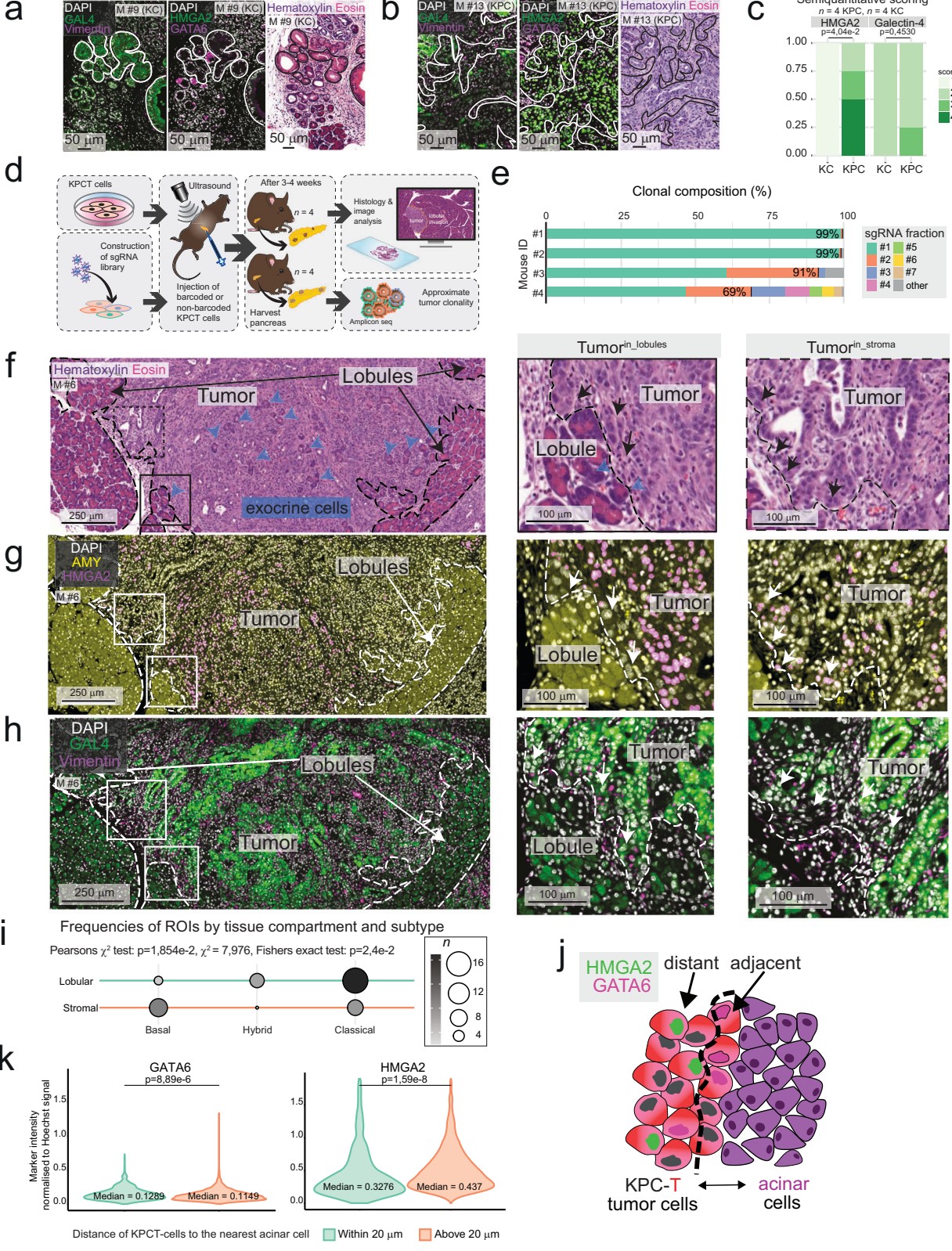

**Nature Communications**| (2025)16:8307                                                                                    **11**

tumor cells located farther from acinar cells, whereas GATA6 expression was elevated in tumor cells in close proximity to acinar cell nests, although effect sizes were limited. These findings support a model in which the adjacency of acinar/ADM cells to tumor cells promotes, but does not induce a complete shift from a basal to a classical-like phenotype in adjacent PDAC cells (Fig. 6k and Supplementary Fig. 20c).

## Discussion

Here, we demonstrate that tumor colonization of pancreatic lobules represents a significant route of invasion in PDAC. We identify a distinct injury-associated microenvironment within invaded lobules, characterized by ADM and NGFR[+]/PDGFRα[+] inflammation-associated stromal cells. This lobular niche influences tumor cell phenotypes, contributing to spatial heterogeneity within the tumor microenvironment.

**Fig. 6 | Compartment-dependent phenotypic switch in murine tumors in vivo. a, b** Representative immunofluorescence (IF) of Galectin-4 (GAL4; classical subtype marker, green, left), Vimentin (basal/mesenchymal marker, magenta, left), high mobility group AT-hook 2 (HMGA2, basal subtype marker, green middle) and GATA binding protein 6 (GATA6, classical subtype marker, magenta, middle), and hematoxylin and eosin stain (H&E, right) in pancreatic intraepithelial neoplasia (PanIN) in the KC (**a**) and PDAC in the KPC (**b**) mouse models. **c** Stacked bar charts displaying the semiquantitative scorings of PanINs in KC KPC mice, using the mean score of all respective lesions for each mouse. **d** Schematic of the orthotopic injection model. KPCT cells were injected into the pancreata of wild-type C57BL/6J mice. A clonality screen was performed by constructing a single guide (sg)RNA library of the KPCT cells before injection, 10e⁵ barcoded cells. **e** Fractions of unique clones identified in the harvested tumors per mouse. **f–h** Representative H&E (**f**) and IF of HMGA2 (pink, nuclear) and α-amylase (AMY, yellow, cytoplasmic) (**g**), and

GAL4 and Vimentin (**h**) on consecutive sections. Black/white arrows: tumor cells in contact with exocrine cells (middle panel) or stroma (right panel), blue arrows: exocrine cells, dashed line separates tumor, lobules, and stroma. **i** Balloonplot visualizing the contingency table of regions of interest (ROIs) in respective compartments. ROIs (n = 40) were classified into a predominant subtype depending on HMGA2 and GAL4 expression. Two-sided Pearson's Chi-squared test and two-sided Fisher's exact test. **j** Schematic of cocultures. **k** Violin plots of nuclear intensity of the respective markers normalized to nuclear erstain for each tumor cell in one representative well from acinar/KPCT cocultures (n = 48478 cells) showing KPCT in vicinity of acinar cells ("Within 20 μm", green, n = 9270) vs. further from acinar cells ("Above 20 μm", orange, n = 39208). **a–h** Data from n = 4 mice for each model. **c, k** Unpaired two-tailed Wilcoxon rank sum test with Benjamini–Hochberg (BH) correction. BH-corrected p-values are given. Source data are provided in the Source Data file.

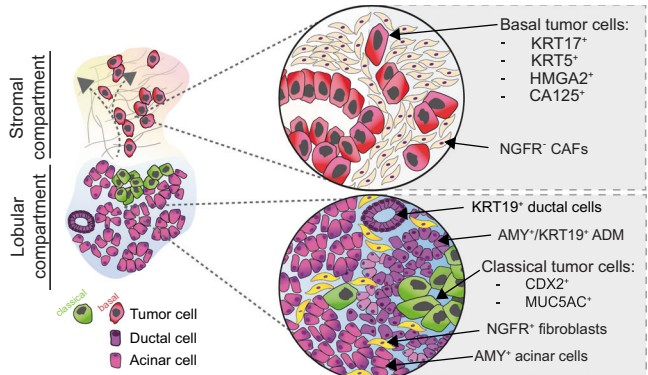

**Fig. 7 | PDAC invasion into an injured lobular microenvironment is linked to a shift towards classical tumor phenotypes.** In the archetypical stromal tumor compartment, where no lobular parenchymal structures remain (upper panel), tumor cells have a predominantly basal phenotype and localize to nerve growth factor receptor (NGFR)-negative cancer-associated fibroblasts (CAFs). In the peritumoral pancreas, the parenchyma is characteristically affected by chronic pancreatitis, in which the pancreatic lobules show varying degrees of atrophy. Lobular stromal NGFR expression is high in these inflamed regions, and some acinar cells undergo acinar-to-ductal metaplasia (ADM), a regenerative response to injury. When the PDAC tumor cells invade the injured lobules, they shift towards a more classical phenotype, and are frequently located next to ADM and acinar epithelial cells (Keratin (KRT)19⁺ α-amylase (AMY)⁺), which are absent in the stroma. Basal markers, KRT17, KRT5, high mobility group AT-hook 2 (HMGA2), carbohydrate antigen (CA)125. Classical markers, caudal type homeobox 2 (CDX2) and mucin 5AC oligomeric mucus/gel forming (MUC5AC).

Reciprocal interactions of tumor cells and their neighboring cells regulate central functions of the PDAC microenvironment, such as tumor cell differentiation and immune tolerance[13,31]. Given that modulation of stromal signaling can profoundly influence tumor progression, a deeper understanding of cellular interactions in distinct tumor microenvironments is warranted[4,5,52]. In the current model, PDAC is characteristically associated with a dense stromal reaction, which envelops the bulk of the tumor cells[53,54]. In addition to stromal invasion, neurotropism and vascular invasion are important anatomic trajectories of PDAC cells[55,56]. Our data quantitatively confirm the abundance of contacts between tumor cells and the fibrotic stroma, including the extracellular matrix, vasculature, and neural elements. However, in addition to the known stromal, perineural, and perivascular niches hosting PDAC cells, our study identifies pancreatic lobules as a significant microniche harboring PDAC cells (schematically depicted in Fig. 7). Pancreatic lobules, the epithelium-rich structural units of the parenchyma, represent a unique microanatomical niche. Particularly at the leading edge of lobular invasion, where PDAC cells

engage with non-malignant epithelial cells and inflammatory fibroblasts, this microenvironment differs from the archetypical stroma, likely exposing tumor cells to a signaling environment distinct from that of the desmoplastic stroma, nerves, or vasculature[23].

In the lobular compartment, we identified benign acinar epithelial cells—at least partially undergoing ADM—as the primary, and exclusively lobular, neighboring cell type adjacent to tumor cells. ADM represents a key component of the reparative response to pancreatic injury, for example, that occurring in pancreatitis[57]. In the context of PDAC, ADM has mainly been considered an early event during tumorigenesis, such that reprogramming of acinar cells is thought to be a major initiator event in cancer formation[21]. This model is supported by data from genetically engineered mice, which indicate that oncogene activation in acinar cells first leads to ADM, followed by the development of PanIN and, finally, PDAC[18,57,58]. Recent scRNA-seq studies have identified ADM cell signatures within PDAC regions, and these findings have been interpreted as that ADM cells act as local precursors of cancer cells, forming the basis for an intratumoral phylogenetic trajectory from ADM to malignant epithelial cells[15]. Our results offer an additional perspective on these data, demonstrating that invading tumor cells within lobules are spatially associated with ADM in the adjacent exocrine compartment. Thus, the presence of ADM gene signatures in advanced PDAC could largely reflect spatial proximity rather than direct lineage derivation. Importantly, this interpretation remains compatible with ADM contributing to the early stages of neoplastic transformation[59].

We find PDAC colonization of the lobules associated with a specific, NGFR⁺/PDGFRα⁺ stroma, whereas NGFR⁺ stromal cells are largely absent in desmoplastic regions. NGFR has been proposed as a marker of benign fibrosis in the pancreas[39] and is a marker of benign fibrosis in the liver[19,38]. Pancreatic stellate cells have been reported to transiently upregulate NGFR when co-cultured with cancer cells, and NGFR⁺ cells are predominantly found in benign lesions such as PanIN, rather than in direct association with the bulk of malignant cells[35]. Together with our findings, these data support a model in which lobular injury, associated with tumor growth, induces a transient NGFR⁺/PDGFRα⁺ stromal phenotype in the peritumoral region, distinct from the established cancer-associated fibroblast subtypes characteristic of the desmoplastic stroma. Alongside ADM, NGFR⁺/PDGFRα⁺ stromal cells delineate a remodeled lobular niche characterized by injury and chronic pancreatitis-like features, possibly providing a distinct microenvironment modifying tumor cell behaviour. The gradual loss of NGFR expression during lobular invasion argues for a model in which the lobular stroma progressively transforms into desmoplastic stroma. However, further studies are required to determine whether this reflects true fibroblast lineage progression from lobular to desmoplastic stromal subtypes. Interestingly, immunomodulatory functions of stromal NGFR have recently been identified in other tissues[60,61]. In light of the pronounced stromal remodeling observed at

the leading edge of PDAC invasion, it will be important to further investigate the associated immune microenvironment in this context. The identification of a lobular niche and its distinct cellular components may open new avenues for targeting microenvironmental regulation of tumor growth. Notably, recent efforts to pharmacologically inhibit ADM represent a potentially promising strategy in this context[62].

We have previously shown that PDAC cells invading the duodenal wall shift from a basal phenotype in the duodenal submucosa (a stroma-rich compartment) to a classical phenotype upon invasion into the epithelial mucosa[46]. Here, we further link the classical phenotype to the proximity to resident epithelial (acinar) cells—a pattern reminiscent of PDAC growth in the duodenal mucosa[46]. These findings suggest that tumor–epithelial interactions, such as those between PDAC cells and non-malignant epithelial cells undergoing ADM, or between PDAC cells and enterocytes[46], may modulate tumor cell phenotypes. In contrast to the extensively studied tumor–stroma and tumor–immune interactions, tumor–epithelial cell interactions remain underexplored. Emerging observations from tumor-alveolar cell interactions in lung cancer and tumor-hepatocyte crosstalk in liver tumors[19,63], and those from the present study, support the notion that interactions within epithelium-rich compartments may contribute to shaping tumor cell behavior and potentially influence therapeutic responses in a tissue context–dependent manner.

While such microenvironmental factors contribute to tumor cell behavior, tumor cell intrinsic, genetic factors drive major phenotypic shifts[32]; specifically, mutations in *KRAS*, *TP53*, and *ARID1A* are enriched in basal tumors, whereas genetic alterations in *SMAD4* are more frequently observed in the classical subtype[33]. It will be important to understand how specific genetic and microenvironmental cues interact to shape the final tumor phenotype. This multifactorial model underscores the complexity of PDAC heterogeneity and the need to integrate both intrinsic genetic factors and extrinsic tissue compartmentalization when delineating and targeting PDAC subtypes.

Based on our findings, we propose a model in which PDAC invasion into pancreatic lobules is accompanied by a transient, injury-associated microenvironment that includes ADM and NGFR⁺/PDGFRα⁺ stromal cells. This lobular microniche may contribute to shaping tumor cell phenotypes during early invasion, prior to the establishment of a fully desmoplastic stroma. A better understanding of such epithelial-rich, injury-associated compartments could have important implications for early PDAC detection and intervention. Specifically, recognizing the cellular features of lobular colonization, such as ADM and the transient NGFR⁺ stroma, may inform diagnostic strategies aimed at identifying tumors at a stage before desmoplasia is fully established. While our study highlights lobular invasion as a previously underappreciated growth pattern in PDAC, the overall contribution of this niche to disease progression and prognosis remains to be fully defined. Additional research is needed to clarify whether tumor cells within the lobular niche exhibit distinct therapeutic vulnerabilities or immune interactions. Ultimately, integrating spatial context, cellular microenvironment, and tumor-intrinsic features may offer a more complete understanding of PDAC heterogeneity and inform strategies for subtype-specific targeting.

## Methods
### Ethics statement
The work on human samples and the download, storage, and analysis of publicly available pseudo-anonymized human sequencing data was approved by the Swedish National Ethical Review Board through the regional ethical committee in Uppsala, Department 2 Medicine (Etikprövningsmyndigheten, #2020-06115, and amendment #2024-06892-02). Informed consent was waived. No compensation was provided for participants. Patient characteristics can be found in Table 1. No sub-analyses were performed based on sex or gender due to the limited sample size. The animal experiments were approved by the Swedish Board of Agriculture through the regional ethics committees Linköpings djurförsöksetiska nämnd, (#00217-2022, injection model) and Stockholms Södra djurförsöksetiska nämnd (#S66-14, KC model and #S31-155, KPC model.). The maximum tumor size permitted was 1500 mm³, which was not exceeded.

### Patients and slide selection
Patients who underwent surgical resection for PDAC in 2017 and 2020 were identified in electronic databases at Karolinska University Hospital, Huddinge, Sweden (Supplementary Fig. 1). In clinical routine, one section from each tissue block from the gross sectioning is subjected to H&E staining. Based on the tissue content in the H&E sections, at least one block is chosen by the pathologist diagnosing the case and further sectioned and labelled with a comprehensive IHC panel. Consecutive slides from cases for which (1) IHC had been performed during clinical diagnosis, and (2) sufficient tumor cellularity allowed quantification in lobular and stromal compartments, were selected for the IHC study. WSIs were obtained by scanning with a Hamamatsu NanoZoomer S360 digital slide scanner at 40× magnification.

### Assessment of the presence of lobular invasion
All available physical glass slides generated in the diagnostic workflow at the Department of Clinical Pathology and Cancer Diagnostics, Karolinska University Hospital, Huddinge, Sweden from $n = 114$ individual PDAC resections (years 2017 and 2020) were retrieved from the archive. 108 cases were sufficiently sampled to assess lobular invasion (Supplementary Fig. 1). All slides were reviewed to identify whether tumor cells were present within pancreatic lobules or not, verified by clinical pathologists.

**Immunohistochemistry.** Tissue fixation, dehydration, paraffin embedding, sectioning, and IHC were performed as part of the standardized diagnostic workflow at the Department of Clinical Pathology and Cancer Diagnostics, Karolinska University Hospital, Huddinge, Sweden. Briefly, single and multiplex IHC was performed on 4–5 µm thick sections on a Leica BOND-III automated staining machine. Antibodies and (pre-)treatment conditions are given in Supplementary Table 1.

### Digital scoring of tumor localization
Using both H&E and p53 IHC WSIs of $n = 5$ tumors, the area of all tumor cells was annotated with the "Brush" tool in QuPath[28] version 0.4.3. The tumors were selected to (1) represent different degrees of acinar atrophy in areas of lobular invasion, and (2) match the selection of cases included in the ML algorithm workflow described below. All annotations were classified as "lobular", "stromal", or "other". The class "other" was used for tumor located in the common bile duct, pancreatic ducts, adipose tissue, nerves, immune cell clusters, or vessels. The areas for each class were then exported for downstream analysis.

### Digital quantification of compartment-dependent heterogeneity
Up to $n = 18$ ROIs were identified on $n = 31$ tumors (see Table 1 for patient characteristics) by a pathologist on H&E, blinded to the other IHC stains (A.Z.). The ROIs were then blindly assigned to either tumor$^{in\_lobules}$ (up to $n = 9$) or tumor$^{in\_stroma}$ (up to $n = 9$). Tumor$^{in\_lobules}$ was defined as tumor cells inside a pancreatic lobule, where lobular cell types or remnants, such as endocrine islets of Langerhans, were present. Subsequently, the corresponding regions were identified in the consecutive WSIs ($n = 159$ in total) of the IHC for quantitation and annotated in QuPath. Annotations were done using the polygon tool, including only tumor cells in the given compartment, until they included between 299–864 tumor cells/ROI (Supplementary Fig. 3a). If

the blindly defined regions initially marked were no longer present for quantitation due to compartment differences between consecutive sections, tissue ruptures or sparse tumor cell numbers, ROIs were moved ($n = 99$) or expanded ($n = 32$) from the initially set region to the nearest compartment-matched region in the section (Supplementary Data 1). For the same reasons, ROIs across IHC within the same tumor could be moved ($n = 14$) or expanded ($n = 26$). Next, cell detection and thresholds for the channels corresponding to each marker were applied. The parameters for cell detection and thresholds were controlled and determined for each WSI by visual assessment (Supplementary Data 1, Supplementary Table 2). QuPath measurements were finally exported as .csv files, annotation-wise, for downstream analysis in R Studio, publicly availably at Github (see "Code Availability").

### Generation of supervised machine learning algorithms for quantification of interactions between tumor cells and non-malignant host cells in lobular and stromal compartments

A nested convolutional neural network-based model was developed with Aiforia Create v 5.5[64] to identify the different cell and tissue components involved in lobular and stromal invasion on digital WSIs of human PDAC ($n = 5$) marked immunohistochemically for CD146 and NGFR (Supplementary Fig. 11a). In brief, within the platform, the user defines "ground truth" by delineating histological structures manually in digital WSIs and assigning them to the user-defined classes. Aiforia uses this information to train the model and compares the model prediction to manually assigned regions. Here, the model was primarily trained to recognize the main structures of the pancreas i.e., acinar cells, ducts, acinar cells undergoing ADM, fibroblasts and extracellular matrix (considered together as "fibrosis"), vessels, nerves, immune cell clusters, endocrine islets of Langerhans, and tumor cells. First, manual annotations (ground truth) capturing the size and morphological variation were assigned to increase the robustness of the model. Subsequently, a model was trained, and the output was visually evaluated using Aiforia's verification and analysis features. Further training annotations were added for structures that were challenging for the initial model. The process was iterated until the model's performance was deemed adequate based on visual evaluation by a specialist pathologist, and based on verification metrics, with no significant changes in performance during the final training rounds. The final model incorporated a total of $n = 664$ training regions and was trained over $n = 3000$ iterations. A detailed description of the training procedure has been given earlier[65] and instructions on how to use the platform are accessible online (www.community.aiforia.com). Details on model hyperparameters and error metrics that Aiforia provides are given in Supplementary Table 3.

### Selection of regions of interest and ML model evaluation

Manually delineated ROIs for different stages of lobular invasion ($n = 15$) and desmoplastic stroma ($n = 26$), were analyzed using the developed ML model for tissue segmentation and classification. Accuracy of results was assessed visually by a specialized pancreas pathologist. The resulting ML-based segmentations were migrated from Aiforia API v 1[66] into QuPath[28], and visually curated by manually correcting for major misclassifications, for example, in areas where tumor and endocrine islets of Langerhans or ducts were misclassified (Supplementary Fig. 11a, **box 2**). After correction, the model accuracy was validated against manual annotations of the tumor cell contacts with the same tissue classes that the model was trained on for lobular invasion ($n = 3$ ROIs) and tumor in stroma ($n = 5$ ROIs) in one WSI of human PDAC. If the contacting tissue with tumor cells was unidentifiable to the predetermined classes above due to challenging morphology or sectioning/stain artefacts, the contact was assigned to class "other". After validation, the invasion front of each lobular ROI was annotated, and cell detection was performed for each tumor, ADM, and acinar annotation in QuPath. Next, the spatial distances

between the detected cells and the corresponding ML-based annotations, together with annotations of invasion front, were computed in QuPath. This was done using QuPath's built-in command "Distance to annotations 2D" that provides the closest distances between all annotations and cell detections, allowing measuring of the distance of each tumor cell, ADM cell, or acinar cell to each ML-identified tissue class. Class annotations (e.g., ADM, fibrosis) within 20 μm from the tumor cell detection centroid were considered as contacts, or tumor adjacent, since the centroid of tumor cells does not directly interface with the surrounding tissue (Supplementary Fig. 13a).

### Mice

For orthotopic injections, C57BL/6J female mice of 9–11 weeks of age were housed in specific-pathogen-free conditions at 12 h light/dark cycle (M #1–8). Mice had *ad libitum* access to standard chow (Mucedola, #2918) and water, and were housed at 20–22 °C. Unstained sections from four 6 month old female KC (KrasLSL-G12D;*Pdx-1-Cre*, M #9-12) mice, and from five KPC (*KrasLSL-G12D/+;Trp53LSL-R172H/+;Pdx-Cre*) mice previously described[67] were used. KPC mice were euthanized at 2–5 months of age. One KPC mouse had no invasive tumor and was excluded from analysis, leaving four KPC mice with similar degree of invasive tumor expansion of the pancreas for semiquantitative analysis (one female, three male, M #13–16). Due to the limited sample size, sex was not considered in the study design for sub-analysis.

### Ultrasound-guided orthotopic injection

For orthotopic injection, an in-house murine PDAC cell line was used ("KPCT"). Briefly, the cell line was produced as follows: KPC mice were bred to *B6.Cg-Gt(ROSA)26Sortm9(CAG-tdTomato)Hze/J* mice to generate KPCT mice. Dissociated pieces of KPCT adenocarcinoma from the KPCT mice were provided by Rainer Heuchel (Department of Clinical Science, Intervention and Technology, Karolinska Institutet, Stockholm, Sweden). The KPCT cell line was cultured in DMEM/F12 medium (Gibco) with 10% fetal bovine serum (FBS; Sigma-Aldrich) and 1% Penicillin-Streptomycin (Sigma-Aldrich) at 37 °C in 5% $CO_2$. At injection day 0, the KPCT cells were in passage 18. The orthotopic injection of KPCT cells into the pancreata of mice was performed under anesthesia using ultrasound-guided identification of the pancreas with the VEVO 3100 preclinical imaging system (Visualsonics, Toronto, Canada). Mice were anesthetized using isoflurane inhalation. The pain killer, buprenorphine, was administered subcutaneously at 0.05 mg/kg before the injection. The fur on the abdomen and left lateral side was gently removed. The mouse was placed in supine position on pre-warmed table in the imaging system, and the pancreas was located with ultrasound. KPCT cells (500,000 cells in max 50 μl sterile phosphate-buffered saline (PBS) were injected into the pancreas with a 30 G needle under ultrasound guidance. Mice were euthanized on day 18 ($n = 1$, M #7), or day 21 ($n = 3$, M #5, M #6, M #8). The humane endpoint was defined depending on the welfare assessment using a combined score that included the animals' weight, body posture, physical appearance, behavior, and urine and fecal excretions and welfare assessment led to euthanasia of M #7.

### Tissue processing

After euthanasia, the pancreata were harvested, briefly washed in PBS and placed in 4% paraformaldehyde (Sigma-Aldrich, #1004965000). After 24 h fixation, the tissue was placed in 70% EtOH for at least 24 h followed by paraffin-embedding and sectioning at 4–5 μm thickness. For next-generation sequencing (NGS) library preparation, pancreata were harvested, frozen on dry ice and stored at −80 °C until further processed.

### Generation of barcoded murine pancreatic adenocarcinoma cells

An existing single guide RNA library[68] cloned by Gibson assembly into pLenti-Puro-AU-flip-3 × BsmBI (Addgene, #196709)[69] and packaged into lentivirus in HEK-293T (ATCC) using plasmids psPAX2 (a gift from

Didier Trono, Addgene, #12260) and pCMV-VSV-G (a gift from Bob Weinberg, Addgene, #8454) was used to barcode cells. Briefly, functional titer was estimated from the fraction of surviving KPCT cells after transduction with different amounts of virus and puromycin selection. Next, KPCT cells were transduced with the library virus at an approximate multiplicity of infection of 0.3 in the presence of 2 μg/ml polybrene. Transduced cells were selected with 2 μg/ml puromycin from day 2 to day 6 post-transduction. For subsequent analysis of genomic DNA, NGS library preparation, and NGS, tumors were homogenized in liquid nitrogen using Freezer/Mill® 6870 (SPEX-sample-prep, 6 cycles, rate 14, 2 min on, 2 min off). The ground tissue was resuspended in Tail buffer (100 mM Tris, 5 mM EDTA, 0.2% SDS, 300 mM NaCl at pH = 8). Genomic DNA was isolated as described previously[70] with the following modifications: QIAGEN Protease (QIAGEN, #19157) and PureLink RNAse A (Invitrogen, #12091021) were used for protein and RNA digest, respectively, and the amount of pre-chilled 7.5 M ammonium acetate (Sigma, #A1542) used for protein precipitation was scaled accordingly. Barcode-containing amplicons were created, sequenced and analyzed as described[69] using the modified primers:

PCR2_FW

acactctttccctacacgacgctcttccgatctcttgtggaaaggacgaaacac

PCR3_fw

aatgatacggcgaccaccgagatctacacacactctttccctacacgacgctct.

The amplicon was sequenced on Illumina NovaSeq6000, reading 20 cycles Read 1 with custom primer CGATCTCTTGTGGAAAGGAC-GAAACACCG. NGS data was analyzed with the MaGeCK software v.0.5.6[71]. Initial barcode complexity was confirmed by including a sample of 40 M cells harvested five days after lentiviral barcoding. 77,361 of theoretically 77,441 barcodes were detected. Read counts of all tumor samples were normalized to total read count and rounded to integers. An arbitrary read count threshold of 1000 was set, and all barcodes with counts below this threshold were removed (in each sample, the removed barcodes contributed less than 0.3% of total reads). Some samples were clonal or near clonal, and in these samples, one or very few barcodes have the overwhelming majority of the read counts, which caused these to bleed through to all other samples due to index hopping, a phenomenon occurring on patterned Illumina flowcells. After filtering for this effect, 931 of 943 barcodes were present in only one of the samples, demonstrating that the samples are independent of each other. Frequencies at which each guide contributes to a sample were calculated based on the normalized total sample read count before filtering.

### Immunofluorescence of murine and human PDAC FFPE sections

Formalin-fixed, paraffin-embedded (FFPE) sections were baked for 1 h at 60 °C after which they were deparaffinized, rehydrated, and rinsed in distilled water. Antigen retrieval was performed with pressure kettle (AptumBiologics, #2100) in DIVA decloaker (BioCare Medical, #DV2004MX). After blocking with 1% bovine serum albumin (BSA; Merck, #A9418-100G), 10% goat (Sigma, #G9023)−or donkey serum (Sigma, #D9663), depending on host of secondary antibody, and 0.05% Triton ×-100 (Sigma, #T8787-100 ml) for 1 h at room temperature (RT), the sections were incubated with primary antibody at 4 °C overnight, washed for 3 × 10 min at RT on shaker. Next, secondary antibodies were applied for 1 h at RT followed by washing for 3 × 5 min at RT on shaker. Alternatively, sections were directly labeled with primary antibody for 2 h at RT. All antibodies were diluted to the working concentration with blocking solution. All washing steps were performed with Tris-buffered saline (Bio-Rad, #1706435) with 0.05% Tween-20 (Pan Reac Applichem/ITW reagents #A4974,0250). The sections with directly labeled primary antibody were fixed in 4% formaldehyde for 15 min at RT and additional washing step with PBS before and after fixation and were included. Primary antibodies included HMGA2 (Cell Signaling, #8179, 1:200), AMY (Sigma-Aldrich,

#A8273, 1:100), KRT19 (1:20, Progen, #61010, 1:20), GAL4 (Invitrogen, #PA5-34913, 1:400), Vimentin (Santa Cruz Biotechnology, #sc7557, 1:50), GATA6 (R&D Systems, #AF1700, 1:80), SOX9 (Sigma, #Ab5535, 1:100) and Red fluorescent protein/tdTomato (Nordic Biosite, #ASJ-JJLIOE-150, 1:100). Alexa Fluor 647 goat anti-rabbit IgG (Invitrogen, #A-21245) and Alexa Flour 488 goat anti-mouse IgG (Invitrogen, #A1101, for human PDAC), Alexa Fluor 488 donkey anti-rabbit IgG (Invitrogen, #A1206, for murine PDAC) and Alexa Fluor 647 donkey anti-goat IgG (Invitrogen, #A21447, for murine PDAC) were used as secondary antibodies, all in 1:400 dilution. To directly label the primary antibody for murine PDAC, the Zenon Rabbit IgG Labeling Kit was used according to the manufacturer's instructions (Invitrogen, #Z25302). Nuclear counterstaining was performed with DAPI (Thermo Fisher Scientific, #121101, 1 μg/ml) for 5–15 min followed by washing at RT. The sections were mounted in ProLong Gold antifade reagent (Invitrogen, #P36924) or Aqua Poly/Mount (Polysciences, #18606). Images were captured using an inverted confocal Nikon Ti2 microscope in widefield mode with a 20× Plan Apo air objective, coupled to a Kinetics sCMOS camera (Supplementary Tables 4 and 5).

### Semiquantitative assessment of HMGA2[+] and GAL4[+] lesions in KC and KPC mice

Upon IF of GAL4 co-labelled with Vimentin, and HMGA2 co-labelled with GATA6, the distribution of GAL4[+] and HMGA2[+] cells was determined by S.SÖ. and C.F.M. jointly. The semiquantitative five-grade scale ranged from 0 to 4, where "0" represented no cells positive, and "4" represented all or almost all cells positive (see code in GitHub for details). For KC mice, the cells of interest that were considered for scoring were exclusively PanINs, while cells of interest for KPC mice were exclusively tumor cells. Each lesion was scored, and a mean score was calculated for each mouse.

### Multiplex-Immunofluorescence

The m-IF labelling was conducted on the Bond RX_m autostainer (Leica Biosystems) as described previously[19]. In brief, deparaffinization was achieved with Dewax solution (Leica Bond, #AR9222) followed by an initial antigen retrieval using Epitope retrieval solution 2 (Leica Bond ER2, #AR9640) at 95 °C for 30 min. Six target cycles were performed in the following antibody order: For the stromal panel, anti-CD74, anti-PDGFRα, anti-NGFR, anti-p53, anti-ASMA, and anti-Vimentin. Details of antibodies and dilutions are provided in Supplementary Table 6. Each cycle included a 5 min blocking step with serum free Protein Block (Agilent, #X0909), primary antibody incubation for 30 min, followed by a 10 min incubation with secondary antibody (either ImmPress-mouse horseradish peroxidase (HRP), MP-7402, or ImmPress-rabbit HRP, MP-7401, Vector Laboratories) and fluorescent labelling for 10 min using tyramide signal amplification with Opal dyes in the sequence of 690, 620, 520, 570, 480 and TSA-DIG/780 (all Akoya Biosciences), diluted 1:300 in 1× Plus Automation Amplification Diluent (Akoya Biosciences, #FP1609). All steps were carried out at RT. Each cycle concluded with an antigen retrieval step with Epitope retrieval solution 1 (Leica Bond ER1, #AR9961) at 95 °C for 20 min. Finally, a 5 min DAPI (Akoya Biosciences, #FP1490) nuclear counterstaining was performed before mounting with ProLong Diamond Antifade media (Thermo Fisher Scientific, #P36970). Each step was followed by subsequent washing steps with Wash solution (Leica Bond, #AR9590).

### Quantification of HMGA2[+] and GAL4[+] murine tumor cells in the injection model

On sections of murine PDAC (n = 4) with HMGA2 and AMY IF co-labelling, ROIs of stromal invasion (n = 16) and lobular growth (n = 24) were delineated. The definition of the ROIs was done by one researcher on H&E-stained sections, blinded to IF. Two other researchers (A.V., S.SÖ.) performed the quantification of the matched IF images. The

HMGA2 and GAL4 channels were kept off while delineating the corresponding IF regions. For each ROI, cell detection based on DAPI staining and positive cell detection based on mean nuclear intensity for HMGA2 and mean cellular intensity for GAL4 were performed using QuPath. Positive cells were called by visual assessment. An example of positive cell detection is given in Supplementary Fig. 19a. Quantitation data were exported in tabular format for downstream analysis.

## Quantification of KRT19⁺ and AMY⁺ cells in human PDAC

All lobules present on the sections ($n = 113$ from $n = 8$ individual tumors) were annotated in QuPath followed by cell detection, based on the nuclear DAPI staining on the KRT19_AMY image. Tumor cells were identified by morphology and p53⁺, as seen on the consecutive section with p53 immunolabeling, and were annotated and removed from the cellular count if present within a pancreatic lobule. Cellular classifiers were trained to identify the following cell classes; "stroma", "periductal stroma", "stroma with α-amylase leakage", "ductal; single positive for KRT19", "ductal; with AMY remnant in lumen", "endocrine cell in islet of Langerhans", "immune", "exocrine; single positive for AMY" and "exocrine; double positive for KRT19 and AMY." Data was exported to R annotation-wise, where fractions of the cell types "ductal; single positive for KRT19", "exocrine; single positive for AMY" and "exocrine; double positive for KRT19 and AMY" were calculated out of the total cell count of each lobule. The lobules were staged to one out of five categories of atrophy, blinded to the results of the cellular classifiers, considering H&E and CD146_NGFR IHC from consecutive sections as: minimal/absent ($n = 7$), mild ($n = 17$), moderate ($n = 48$), severe ($n = 28$) and end-stage ($n = 13$) atrophy, in line with scoring guidelines for chronic pancreatitis[72]. In short, the degree of acinar cell loss, fibrosis, fatty infiltration, and duct changes, such as distortion and dilatation, was used to categorize atrophy severity. Assessments were supervised by a pancreatic pathologist (C.F.M.).

**IHC-based quantification of stromal stains.** For human PDAC sections ($n = 7$) double-labelled with IHC for CD146 (brown chromogen) and NGFR (red chromogen), NGFR positive area of $n = 67$ lobules was quantified. The lobules were categorized as: unaffected ($n = 16$), for the most normal pancreatic lobules on the sections (invasion-, atrophy-); inflamed ($n = 20$) with signs of chronic pancreatitis but no tumor invasion (invasion −, atrophy +); early invasion ($n = 16$), i.e., slightly invaded lobules with pancreatitis (invasion +, atrophy ++); and late invasion ($n = 15$), i.e., extensively invaded lobules with remnants of lobular structures, such as islets of Langerhans and ADM (invasion ++, atrophy +++). A high-resolution (0.92 μm/pixel) pixel thresholder for the red NGFR channel was applied to each ROI with customized cutoffs for each section based on visual evaluation in QuPath to account for intensity variations (Supplementary Fig. 13c). For sections with ML-based annotations, excluding case 7 ($n = 4$) due to end-stage morphology and negativity for NGFR, NGFR intensities of ROIs for lobular invasion ($n = 12$ ROIs) and desmoplasia ($n = 20$ ROIs) were analyzed in QuPath. For each cell detection inside ML-derived acinar, ADM, and tumor annotations, the mean intensity of the red NGFR channel based on 0.1 μm pixel size inside a surrounding circular area (diameter 50 μm) was measured (Supplementary Fig. 13a **boxes 2 and 3**). Only cell detections directly interacting with ML-derived fibrosis annotations (i.e., distance from cell detection centroid to the closest fibrosis annotation was less than 20 μm) were included for downstream analysis. All negative intensity values (a computational artefact) were set to zero.

**m-IF based quantification of stromal markers**
Tumor clusters were delineated with the polygon tool in QuPath, using p53⁺ nuclei for tumor identification. For each case, $n = 6$ annotations for tumor^in_lobules and $n = 7–10$ annotations for tumor^in_stroma were drawn, resulting in $n = 47$ ROIs for tumor^in_lobules and $n = 74$ ROIs for

tumor^in_stroma. Three area layers expanded around to the tumor annotation were defined at 15 μm increments (Supplementary Fig. 14a). To analyze the stromal cells present in the different compartments, larger lobular areas ($n = 53$) or stromal areas ($n = 40$) were annotated (Supplementary Fig. 15a, b). Cell detection was performed within the annotations with subsequent removal of cells with high nuclear circularity to enrich for stromal cell content. A cell classifier was trained to recognize cells that were positive or negative by visual assessment and applied to all cell detections. In the subsequent data analysis, any non-stromal residual cells were removed by excluding cells detected as p53⁺ by the classifier.

## Analysis of scRNA-seq data

**Data preprocessing and quality control.** Previously published data sets on mouse PDAC progression (GSE217846) or pancreatic injury (GSE250486) and on human PDAC (GSE194247, CRA001160, Supplementary Table 7) were re-analyzed for this study with Seurat v 5.1.0. Quality control filters for mitochondrial gene expression (Caronni et al.: <25%; the rest of the analysed data sets: <10%), feature counts (Peng et al. & Kim et al.: > 500 & < 7000; Falvo et al.: > 800 & < 4000; Caronni et al.: >200), and read counts (Peng et al. & Kim et al.: > 2000; Caronni et al.: >1000) were applied. scDblFinder v 1 with dbr set to 0.07 was used to filter out the doublets[73]. Canonical correlation analysis of Seurat with default parameters was applied for sample integration when necessary. Default Seurat pipelines were followed for data normalization, feature calling, and scaling. Dimensionality reduction from multidimensional space was done with shared nearest neighbor and 20 first principal components (PCs) were selected for further analysis. Multidimensional data were reduced into two dimensions for visualization purposes with uniform manifold approximation and projection (UMAP).

## Gene signature analysis

Fibroblasts with NGFR read counts higher than zero were assigned as *NGFR*⁺ subset and specific marker expression was defined with FindMarkes command of Seurat using default parameters and visualized with EnhancedVolcano v1.22.0 (https://github.com/kevinblighe/EnhancedVolcano). The returned gene list was then filtered to include upregulated genes with logFC >1 and adjusted *p*-value < 0.01 to determine a NGFR gene set signature. Gene sets for CAF and PDAC subtype signatures were retrieved from previously reported publications of mouse and human PDAC; Elyada et al. Supplementary Tables S13 and S22[9], Moffit et al.[45]. Supplementary Table 2 and Pitter et al.[44]. Supplementary Table S1. Gene signature scores were computed with UCell v 2.8.0 and the command AddModuleScore_UCell[74].

## Gene enrichment analysis

The filtered NGFR signature gene lists were used to retrieve enriched Gene Ontology terms associated with biological processes using ClusterProfiler v 4.12.6[75] and mapped against genome-wide background annotations of org.Hs.eg.db v 3.19.1 and org.Mm.eg.db v 3.19.1.

## Statistics and reproducibility

All downstream analyses were conducted using R v 4.1.2 and 4.2.2[76]. Dplyr v 1.1.3[77], tidyr v 1.3.0[78], tidyverse v 2.0.0[79], stats v 4.1.2[76], data.table v 1.14.8[80], magrittr v 2.0.3[81], stringr v 1.5.0[82], reshape2 v 1.4.4[83] Hmisc v 4.8-0[84] and forcats v 1.0.0[85] were used for data wrangling. Summary statistics were performed with BioGenerics v 0.40.0[86]. Data from the compartment-dependent heterogeneity and m-IF pipeline were analyzed with Benjamini−Hochberg-corrected, paired or unpaired two-tailed Wilcoxon rank sum test or Kruskal−Wallis test with Dunn's test as a post-hoc using rstatix v 0.7.2[87]. Unpaired two-tailed Wilcoxon rank sum test to compare NGFR positive areas across different lobular atrophy conditions was performed with rstatix v 0.7.2[87]. Visualizations were done with ggalluvial v 0.12.5[88], ggplot2 v 3.4.3[89], ggpubr v 0.6.0[90],

ggforce v 0.4.1[91], ggbreak v 0.1.2[92], ggridges v 0.5.4[93], ggVennDiagram v 1.2.2[94], Biovenn v 1.1.3[95], circlize v 0.4.15[96], ComplexHeatmap v 2.10.0[97], corrplot v 0.92[98], factoextra v 1.0.7[99], ggthemes v 4.2.4[100], hrbrthemes v 0.8.0[101] and Formula v 1.2−5[102] and enhanced using color scales from RColorBrewer v 1.1.3[103], survminer v 0.4.9[104], viridis v 0.6.2[105], viridisLite v 0.4.2[106] and wesanderson v 0.3.7[107]. No data imputation was performed, except for assignment of the value "0" for the percentage of marker-positive tumor cells in ROIs in the compartment-dependent heterogeneity pipeline that were considered completely negative for a given IHC marker. No statistical method was used to predetermine sample size. Where data are presented as Box-and-Whisker plots, the median (line), the interquartile range (IQR, box), minimum and maximum values within 1.5 times IQR from the first and third quartile (whiskers) are shown. Number of ROIs for boxplots visualizing the compartment-dependent heterogeneity for each marker (Supplementary Figs. 4−9) are given in Supplementary Data 1.

**Isolation of acinar cells.** Following euthanasia, pancreata were excised and acinar cells were isolated as previously described[108,109]: Briefly, pancreata were rinsed twice with cold HBSS without $Ca^{2+}$ and $Mg^{2+}$ (Capricorn Scientific) supplemented with 1% Penicillin-Streptomycin (Sigma-Aldrich) and subsequently cut into 1−3 mm³ fragments. The dissected tissue was centrifuged at $450 \times g$ for 2 min at 4 °C, and the resulting pellet was resuspended in 10 mL of pre-warmed collagenase IA solution (HBSS without $Ca^{2+}$ and $Mg^{2+}$, 10 mM HEPES [Gibco], 0.25 mg/mL trypsin inhibitor [Sigma-Aldrich], 200 U/mL collagenase IA [Sigma-Aldrich]). Digestion was performed at 37 °C with shaking at 220 rpm for 20 min. To stop enzymatical digestion, 10 mL of cold buffered washing solution (HBSS without $Ca^{2+}$ and $Mg^{2+}$, 10% FBS [Sigma-Aldrich], 10 mM HEPES) was added. The cell fragments were washed three times with cold buffered washing solution and filtered through a 100 μm mesh. Pancreatic structures passing through the mesh were maintained overnight in complete Waymouth's medium (Waymouth's medium [Merck] supplemented with 2.5% FBS, 1% Penicillin-Streptomycin, 0.25 mg/mL trypsin inhibitor, and 25 ng/mL recombinant murine EGF [Peprotech]). After 24 h, acini were transferred to fresh 12-well cell culture plates coated with 5 μg/cm² collagen I. Acinar structures adhered within two days, and the medium was refreshed every three days for up to ten days. For co-culture experiments, KPCT cells (passage 33) were pre-cultured in complete Waymouth's medium for seven days before co-culture initiation. Three days after acinar cell isolation, 5000 KPCT cells were seeded into each well of a 12-well plate containing the attached acinar cells. Co-cultures were maintained for three days prior to fixation.

**Immunofluorescence of in vitro cultures**
Acinar cells as well as co-cultures of acinar and KPCT cells, were fixed in 4% paraformaldehyde for 15 min. Unless otherwise stated, all steps were performed at RT. Cells were washed three times with PBS, permeabilized with PBS containing 0.1% Triton ×-100 (Sigma, #T8787-100 ml), and blocked for 30 min in PBS supplemented with 5% BSA (Merck, #A9418-100G) and 0.1% Triton ×-100 (Sigma, #T8787-100 ml). For immunolabelling, cells were incubated overnight at 4 °C with primary antibodies GATA6 (R&D Systems, #AF1700, 1:100), in duplex with HMGA2 (Cell Signaling, #8179, 1:100) or α-amylase (Sigma-Aldrich, #A8273, 1:100) as single stain, diluted in antibody dilution buffer (PBS, 0.1% Tween-20 (Pan Reac Applichem/ITW reagents #A4974,0250), 1% BSA (Merck, #A9418-100G)). The next day, cells were washed three times with PBS and incubated for 1 h in dark with secondary antibodies including Alexa Fluor 488 donkey anti-rabbit IgG (Invitrogen, #A1206, 1:400) and Alexa Fluor 647 donkey anti-goat IgG (Invitrogen, #A21447, 1:400), along with Hoechst 33342 (BioTechne, #5117, 1:500) for nuclear counterstain in antibody dilution buffer, and lastly washed in PBS. Images were captured using an inverted confocal Nikon Ti2

microscope in widefield mode with a 20× Plan Apo air objective, coupled to a Kinetics sCMOS camera (Supplementary Table 8).

**Reporting summary**
Further information on research design is available in the Nature Portfolio Reporting Summary linked to this article.

## Data availability
Source data are provided with this paper. The Data generated in this study and a minimum dataset that support the findings of this study are available on GitHub, https://doi.org/10.5281/zenodo.15856093. The single-cell sequencing datasets analysed in this study are available in Gene Expression Omnibus (GEO), https://www.ncbi.nlm.nih.gov/geo/) and the National Genomics Datacenter repository Genome Sequence Archive (https://ngdc.cncb.ac.cn/gsa/browse), respectively, with the accession codes: GSE217846, GSE250486, GSE194247, and CRA001160. The processed clinical and imaging data generated from the human patient cohort are available under restricted access for privacy and legal reasons. Access can be obtained for research purposes by submitting a request to the corresponding author (M.G.) by email, specifying contact details, affiliation, and purpose of the request. All requests for clinical data used in this study are reviewed by the investigators responsible for the clinical cohort (M.G. and C.F.M.), and requests will be answered within four weeks. Any data that can be shared after mandatory review by the National Swedish Ethical Review Board will be released via a Data and/or Material Transfer Agreement to the requesting party; this includes all relevant pseudonymized clinical data (such as relapse and overall survival, information on treatment, as well as the digitally annotated imaging data, including H&E stains and IHC stains with annotations). Authorship requirements for the use of the clinical data from this cohort will be regulated in a Collaboration Agreement, in which the authors agree on authorship requirements prior to data sharing. Processed images of the mouse experiment data and the in vitro experiments are available on GitHub. The raw, unprocessed images, including annotations, are available from the corresponding author upon request. Source data are provided with this paper.

## Code availability
Analyses were performed in R with publicly available packages. The data processing and analysis code is available on GitHub https://doi.org/10.5281/zenodo.15856093.

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

## Acknowledgements
The PhD student position of S.SÖ. is supported by Karolinska Institutet. This study was supported by The Swedish Research Council (projects nr. 2018-02023 and 2024-03026, to M.G.), StratRegen (to M.G.), The Swedish Society for Medical Research (to M.G.), The Swedish Cancer Society (22 2175 Pj, to M.G. and 23 2856 Pj to St. S.), the Center for Innovative Medicine (to M.G. and J.E.), Cancer Research KI (to M.G. and E.S.), and Radiumhemmet's research funds (to M.G.). J.E. was supported by Region Stockholm, the Bengt Ihre Foundation, and Radiumhemmet's research funds. C.F.M. is supported by The Swedish Society for Medical Research (PD21-0114) and Ruth and Richard Julin's foundation. N.G. is supported by the German Research Foundation (#460567311) and Ruth and Richard Julin's foundation. Part of this work was carried out at the SciLifeLab CRISPR Functional Genomics unit at Karolinska Institutet, funded by Science for Life Laboratory, and Live Cell Imaging core facility/Nikon Center of Excellence at Karolinska Institutet, supported by the KI infrastructure council, and the Department of Clinical Pathology and Cancer Diagnostics. The CRISPR Functional Genomics unit acknowledges support from the National Genomics Infrastructure, the Swedish National Infrastructure for Computing (SNIC), and the Uppsala Multidisciplinary Center for Advanced Computational Science (UPPMAX). We thank Kjetil Søreide and Anna Uddén for helpful comments on the manuscript and Andrea del Valle for support with animal experiments. We thank Linda Lindström, Oscar Danielsson, Sólrún Kolbeinsdóttir, Nick Tobin, and Carsten Daub for helpful advice and discussions.

## Author contributions
C.F.M. and M.G. designed and supervised the study together with B.B. A.Z. performed the blinded ROI selection on clinical multiplex IHC sections. S.SÖ. performed the quantification of consecutive multiplex IHC and analyzed the data under supervision of M.G. and C.F.M. S.SÖ. performed the lobular invasion screen and scoring of tumor localization under supervision of K.R., C.F.M., and B.B. P.G., J.E., E.S., C.SA, C.F.M., and M.G. collected clinical data and clinical samples, S.H. and Y.H. curated the clinical sample collection. A.V. trained the supervised machine learning algorithms, quantified and analyzed the data of tumor interactions and NGFR expression under supervision of C.F.M., S.SÖ., and M.G. S.SÖ. and A.V. performed IF on murine samples, and analyzed and interpreted the results with C.F.M. and M.G. O.K., S.M., and B.S. generated the barcoded KPCT cells and analyzed the results together with A.V. C.ST. and N.H. performed and analyzed multiplex-IF. S.SÖ. analyzed the multiplex-IF data under the supervision of N.G., M.G., and C.F.M. A.N., A.V., and S.SÖ. performed, analyzed, and interpreted duplex-IF experiments, for which M.G. and C.F.M. blindly selected ROIs. M.Z. and T.C. generated KC mice and collected their pancreatic tissues under the supervision of S.ST. M.G., N.G., and S.SÖ. performed orthotopic injection experiments, and M.G. S.SÖ. performed confocal microscopy. A.K. and S.SÖ. performed the acinar cell isolations and co-culture experiments together with N.G. S.SÖ. and A.V. wrote the code, generated plots for and prepared figures under the supervision of M.G. M.G. wrote the manuscript together with S.SÖ., A.V., and C.F.M. N.G., K.R., C.ST., J.E., and S.H. edited the manuscript. All authors commented on the manuscript.

## Funding

## Competing interests
The authors declare no competing interests.
