## [Transparent Peer Review file · Nature Communications]

An injury-associated lobular microniche is associated with the classical tumor cell phenotype in pancreatic cancer

Corresponding Author: Dr Marco Gerling

Version 0:

Reviewer comments:

Reviewer #1

(Remarks to the Author)

The study titled “A distinct injury-driven lobular tumor microniche in pancreatic ductal adenocarcinoma” by Söderqvist et al. characterized two PDAC tumor phenotypes (tumor-in-lobule and tumor-in-stroma) and their associated fibroblasts by analyzing multiplex protein/mRNA staining of known markers using machine learning. They also delved into the mechanism that drive the two tumor cell phenotypes using mouse models. It is well-written, and the work is interesting. However, the study will be strongly supported if any single cell dataset recapitulates the two tumor phenotypes and NGFR+/PDGFRα+ stromal cell type.

Specific comments:

Fig 2e. what are the rows and columns represented in the heatmap? Row annotation tracks on the left indicates rows are markers. But why each marker seems to have multiple rows? And what do the arrows mean? Does the arrow pointed up at bottom right with text “ROI replicate” mean the rows are ROI replicates? Then how to interpret the row annotation tracks on the left? Could the authors clarify or revise the plot to avoid confusion.

Fig S4b, is each panel from one patient? i.e. the numbers in panel title are patient IDs?

L178 What other determinants of PDAC subtype identity?

L192 Does ADM have distinct morphology to be clearly identified and annotated? How NGFR was chosen?

L197 The model accuracy should be reported.

Fig 3e Please describe how the degree of lobular atrophy was defined.

Fig 4a Did the author divide the distance from 0 to 500um into bins and summarize cell contacts within each distance bin? The legend mentioned there were 5 tumors and 15 ROIs. How the replicates were used, was mean or median of tumors (or ROIs) taken?

Fig 5b Could the two major clones in mouse #3 enriched in tumor-in-stroma and tumor-in-lobule separately? Is there any way to know if any of the clones associated with specific tumor phenotype.

L317 What evidence would support clonal selection?

L734 The description of how quantification of tumor cell contacts was not clear. “spatial distance between cell detections and ML-based annotations, together with annotations of invasion front, was computed in QuPath”. Could the author explain how cell detections, ML-based annotation and invasion front annotation were used to calculate distance?

(Remarks on code availability)

Reviewer #2

(Remarks to the Author)

In this article, the authors investigated possible cellular interactions in the lobular compartment surrounding PDAC. Using cellular classification through machine learning (ML) on whole slide images of human PDAC and some mouse models, the authors describe specific changes at the interface between PDAC and lobules (Lobular changes: ADM, and NGFR+ fibroblasts; PDAC changes: shift towards classical phenotype) and compare these to features of tumour cells in desmoplastic stroma. Based on this, the authors concluded the existence of “an injury-induced tissue niche” in “nascent PDAC invasion”.

The article is interesting and introduces some novel concepts and techniques. However, there are significant weaknesses (as detailed below) that diminish enthusiasm for the paper in its present form. We also have some concerns regarding overreach of conclusions.

Our specific major comments, in order of importance, are:

1. Assumption that spatial relationships imply temporal relationships lacks convincing experimental support: One of the conclusions of this paper is that the lobular invasion and the features of the cancer/pancreatic lobules in these areas reflect changes in “nascent PDAC invasion”. In other words, the authors are implying these changes reflect the earliest changes in PDAC/pancreas interaction. This conclusion is difficult to justify given the evidence presented in this article represents only a snapshot in time. How can the authors be sure that the tumour cells identified in lobular regions with ADMs are not due to oncogenic changes within existing ADMs in that area? The authors have used the spatial relationships between tissue types as a way to deduce temporal relationships. For firmer conclusions, better experimental data are required either by benchtop or animal studies. For instance, how does PDAC phenotype change with animal models with and without stromal co-injection, or in animal models with various degrees of experimentally induced lobular atrophy?
2. Assumption that cellular proximity implies interaction lacks adequate supportive evidence: There is also a question of whether it is always true that proximity implies “interaction”. While logical, this is not guaranteed. For instance, one can argue that the change in PDAC phenotype in the lobular compartment may be due as much to the loss of signals from stroma as it is from close interaction with lobular cells or ADMs. That is, the interaction may be with the cells far away from the leading edge of invasion rather than with cells directly being invaded. For this reason, we believe that this assumption should be backed up by experimental data which support the existence of such interactions or other collateral evidence of interactions (bioinformatics approach).
3. Validation of ML classification requires strengthening: Given the centrality of the accuracy of the ML classification system to the validity of the study, the authors should provide further data to reassure readers of the accuracy of the ML classification system. This should include both global measures of accuracy, as well as specific measures for specific classes of cells. For instance, Supplementary Figure 11b does not provide any details of the accuracy of any given class of cells, nor a quantified validation of the ML model. This Figure only indicates to the reader that the proportion of fibrosis is much the same between ML and manual annotations. This is because the fibrosis is the major tissue type. But what about other tissue classes (in particular, ADMs)? Also, concordance of overall proportions does not imply accuracy of classification at the cell level, which is more important.
 - o Along the same line, ADM classification accuracy needs clarification. Firstly, the authors have not presented data related to the validation of ML classification of ADM against ground truth. Second, the accuracy of ground truth is called into question – The study only used 2 markers (KRT19 and AMY) to identify ADMs. We note that a more expanded combination of markers (ADM and others) may have been performed (by multiplex ISH), but it seemed like this was not used for ML training nor for validation. Also, there are no data (apart from representative image in Fig 3f) to show that the identified ADMs using limited markers corresponded to identified ADMs using expanded markers (or even negative for non-ADM markers). It would seem appropriate that the most accurate method of ADM identification (probably combination of morphology + multiplex markers) should be used as “ground truth” for ML validation.
4. Mouse model data can be more convincing: The mouse model did not convincingly demonstrate that the changes in the PDAC cells in the lobules is due to environmental cues rather than being clonal. While the tumour diversity (according to the barcoding prior to injection) suggests the dominance of limited clones, there are no data to support or refute that the changes are subclonal (ie, as a subclone of the dominant barcoded clone). The use of only one marker (HMGA2) to indicate PDAC phenotype is also suboptimal.
5. Translational significance is unclear: Finally, the translational implication of these findings is not entirely clear. How do the authors foresee the information from their study being applied to the clinical setting such that it significantly changes patient outcomes? This can be further explored in the discussion.
6. Compliance with SAGER guidelines not articulated: For the animal studies, only female mice have been used. This needs to be mentioned in the abstract as per the journal’s guidelines.

(Remarks on code availability)

I instructed one of my postdocs to access the provided link and can confirm that the data and code have been provided. However to run the code we would need to update our version of the software which is not currently possible because it requires institutional permission.

Reviewer #3

(Remarks to the Author)

In this manuscript Soderqvist and colleagues describe a distinct injury-associated non desmoplastic niche for pancreatic ductal adenocarcinoma cells in pancreatic lobules. They suggest that the tumor invasion front in the lobules is dominated by interactions between tumor and ADM cells and contains injury activated fibroblasts. In this area PDAC cells expressed a more classical phenotype, whereas PDAC cells with a desmoplastic stroma shows more a basal phenotype. This is an interesting finding which could have implications for understanding how PDAC phenotypes occur or interchange; and could have implications for early detection. However, more evidence needs to be included to fully support these points. Macrophages, dependent on their subtype, have been shown to regulate ADM and fibrosis. The subtypes of macrophages in lobular/PDAC and desmoplastic/PDAC are as should be determined and this information should be included into the manuscript.

Figure 3F should be accompanied by a H&E staining. Why are there 3 types of ADM cells indicated by different marker combinations?

The interpretation of Figure 4E seems a bit far-fetched. Maybe immunofluorescence for both, NGFR and PDGFalpha, should be shown in addition to Figure 4e (instead of showing this in the Supplemental Figure 15). In Supplemental Figure 15b it is

unclear if both markers overlap in their expression. If the NGFR+/PDGFRalpha+ injury-induced stromal cells define a distinct cell population of CAFs (independent of previously described CAF populations) as suggested, they should be isolated and characterized.

More than one marker should be used for the basal phenotype in Figure 5, and markers for the classical phenotype should also be included.

The interpretation (Figure 6) is that basal tumor cells, when invading into the injured lobules switch to a classical phenotype and closely interact with ADM cells. But this 'switching' is not really demonstrated and there is no mechanistic insight in how this can occur. Can this be shown ex vivo?

It also is not sure which phenotype occurs first, which would be important for early detection (as suggested by the authors). If their findings can be reproduced in KPC (as shown) it should be possible to do a time course of tumor formation in KPC and then determine relative changes between phenotypes or more/less classical/lobular areas.

In Figure 1C the SMAD4 IHC is not convincing.

(Remarks on code availability)

Reviewer #4

(Remarks to the Author)

(Remarks on code availability)

Version 1:

Reviewer comments:

Reviewer #1

(Remarks to the Author)

The authors addressed all my comments.

I have a few additional comments on the revised manuscript:

Figure 2e. The colors used for the row annotation track of marker gene are not well distinguishable, making it hard to see the trend. Alternatively, the author can consider split rows by marker. It also helps to see the heterogeneity across tumors.

L294 The second part of the paragraph is repeated.

Some of the supplemental figures have low resolution, the text is not readable, for example supp fig 15d.

L353, why different markers for classical tumor are used here? CDX2 and MUC5AC were used at the beginning instead of GAL4, GATA6.

L359-362 It is hard to see the trend as stated "GAL4 and HMGA2 were expressed with limited overlap in classical and basal/mesenchymal tumor cells" in Supp Fig 17b. GAL6 and GATA6 are classical markers, and should they overlap with classical signature? Are they part of the Pitter et al classical signature?

(Remarks on code availability)

Reviewer #2

(Remarks to the Author)

The authors have satisfactorily addressed some of the comments but it would be useful if the following remaining issues relating to comments 1,2 and 3a could be further addressed:

Comments 1 and 2

The provided explanation as to why tumour cells in lobular regions are not multifocal de novo tumor seems adequate and the use of tdTomato labelled cancer cells in mouse model is supportive

The comparison between PanIN in KC mice and tumours in KPC mice is useful, but does not necessarily imply the temporal change over time (i.e., lobular compartment being early -> stromal compartment being late; which corresponds to change in phenotype from classical-> basal subtype), especially given PanINs are non-invasive (i.e., comparison between cancer and non-cancer).

The in vitro data provided 6j-k are interesting, but despite the high statistical significance (this is due to the large number of cells analysed), the effect size does not seem large (from 6k: the violin plots of and the small change in the median marker intensity).

This suggests that the phenotype differences between in-lobule and in-stroma cancer is only partly explained by interaction with ADM. Therefore, the language around the implied temporal relationship/causality needs to be toned down

- classical at early lobular interaction -> basal at late stage, in stroma;
- cancer invasion -> lobular atrophy and ADM; and
- ADM causes cancer to be classical in subtype.

Comment 3a

The provided explanation as to why tumour cells in lobular regions are not multifocal de novo tumor seems adequate and the use of tdTomato labelled cancer cells in mouse model is supportive

The comparison between PanIN in KC mice and tumours in KPC mice is useful, but does not necessarily imply the temporal change over time (i.e., lobular compartment being early -> stromal compartment being late; which corresponds to change in phenotype from classical-> basal subtype), especially given PanINs are non-invasive (i.e., comparison between cancer and non-cancer).

The in vitro data provided 6j-k are interesting, but despite the high statistical significance (this is due to the large number of cells analysed), the effect size does not seem large (from 6k: the violin plots of and the small change in the median marker intensity).

This suggests that the phenotype differences between in-lobule and in-stroma cancer is only partly explained by interaction with ADM. Therefore, the language around the implied temporal relationship/causality should be toned down

- cancer invasion -> lobular atrophy and ADM; and
- ADM causes cancer to be classical in subtype.

(Remarks on code availability)

I instructed one of my postdocs to access the provided link and can confirm that the data and code have been provided. However to run the code we would need to update our version of the software which is not currently possible because it requires institutional permission.

Reviewer #4

(Remarks to the Author)

(Remarks on code availability)

Point-by-point answers to the comments:

Reviewer #1 (Remarks to the Author):

The study titled “A distinct injury-driven lobular tumor microniche in pancreatic ductal adenocarcinoma” by Söderqvist et al. characterized two PDAC tumor phenotypes (tumor-in-lobule and tumor-in-stroma) and their associated fibroblasts by analyzing multiplex protein/mRNA staining of known markers using machine learning. They also delved into the mechanism that drive the two tumor cell phenotypes using mouse models. It is well-written, and the work is interesting. However, the study will be strongly supported if any single cell dataset recapitulates the two tumor phenotypes and NGFR⁺/PDGFR α ⁺ stromal cell type.

We thank the reviewer for their positive assessment of our work and for their constructive comments, which have significantly improved the manuscript.

In response to the reviewer’s suggestion, we have conducted analyses of both human and murine single-cell RNA sequencing datasets. The new data are mainly presented in Fig. 4c–e, Fig. 5e–h, and Supplementary Fig. 15d–e. As outlined in our response to the Editor’s comments, these analyses confirm the presence of NGFR⁺/PDGFR α ⁺ stromal cells across multiple contexts: in murine pancreatitis, in early tumors following orthotopic tumor cell injection in mice, and in small human PDACs. In all datasets, NGFR⁺ stromal cells are consistently enriched for inflammatory pathways, supporting our interpretation that this cell population is associated with lobular injury and is part of a distinct, injury-associated tumor microniche in lobular regions.

Specific comments:

Fig 2e. what are the rows and columns represented in the heatmap? Row annotation tracks on the left indicates rows are markers. But why each marker seems to have multiple rows? And what do the arrows mean? Does the arrow pointed up at bottom right with text “ROI replicate” mean the rows are ROI replicates? Then how to interpret the row annotation tracks on the left? Could the authors clarify or revise the plot to avoid confusion.

We thank the reviewer for pointing out potential ambiguities in Fig. 2e. In this figure, each cell represents the fraction of positive tumor cells for a given marker within a specific region of interest (ROI). Each marker (e.g., KRT17, MUC5AC) is represented by 31 individual rows, one per tumor (T#1, T#2, ..., T#31), resulting in multiple rows per marker. The columns correspond to individual ROIs, separated into stromal and lobular compartments.

To enhance clarity, we have removed the arrows that previously indicated the orientation of rows and columns. The label “ROI replicate” has been revised to “ROI number” to avoid the impression that the rows represent technical replicates. Additionally, we have clarified the figure legend.

Fig S4b, is each panel from one patient? i.e. the numbers in panel title are patient IDs?

Yes, each panel represents an individual patient’s tumor, using pseudonymous numbers in accordance with ethical and legal requirements. We have now updated the panel labels to the format (T #tumor_ID) for consistency and have clarified this in the figure legends for all the related

Supplementary Figures 4-9. Consequently, the previous Supplementary Table 9 has been removed, which contained a legend for the corresponding tumor labels.

L178 What other determinants of PDAC subtype identity?

We thank the reviewer for this interesting question. While our study highlights tumor cell localization—specifically, lobular versus stromal—as an important factor influencing PDAC subtype identity, we acknowledge that it is not the sole determinant.

Recent work, including that by Singh *et al.* (Cancer Res. 2024;30(21):4932–4942), has shown that classical and basal phenotypes frequently coexist within individual tumors, with many exhibiting an “intermediate” profile. Their study also identified distinct mutational patterns—such as differing frequencies of *KRAS*, *TP53*, *ARID1A*, and *SMAD4* mutations—between strongly classical and strongly basal tumors, further supporting the role of intrinsic genetic alterations in subtype specification, as previously suggested (Mueller *et al.* Nature, 2018;554:62–68). In this context, tumor localization contributes to PDAC subtype heterogeneity as an ‘extrinsic’ factor, influencing tumor cell behavior through microenvironmental cues. However, it acts in concert with ‘intrinsic’ factors, including genetic and transcriptional programs. We have now more explicitly incorporated these additional determinants into the revised discussion to reflect this multifactorial model of subtype specification (in the text, for example, Ll. 338ff and 494ff in the manuscript version without tracked changes).

L192 Does ADM have distinct morphology to be clearly identified and annotated?

We thank the reviewer for raising this point, which also reviewer #2 touches upon. ADM (acinar-to-ductal metaplasia) does exhibit distinct morphological features on H&E staining, including a loss of zymogen granules and a transition toward duct-like structures that reveal early or more advanced stages of lumen formation. These features allow it to be distinguished morphologically from, for example, PanIN lesions and intralobular ducts. The original definition of ADM is rooted in histomorphological assessment on H&E, as described, for example, by Klöppel and Longnecker (Ann N Y Acad Sci. 1999; doi: 10.1111/j.1749-6632.1999.tb09510.x), where it was initially referred to as “tubular structures”. Hence, the definition of ADM is morphological, and not based on molecular analyses.

We acknowledge that ADM represents a dynamic process, and its precise initiation and endpoint cannot be sharply delineated. Nonetheless, based on the diagnostic expertise of the specialized pancreas pathologists on our team (CFM, BB), we are confident in our ability to distinguish mature acinar cells from those undergoing acinar-to-ductal transformation using established morphological criteria (Esposito *et al.*, Guidelines on the histopathology of chronic pancreatitis, Pancreatology, Volume 20, Issue 4, June 2020, Pages 586-593, <https://doi.org/10.1016/j.pan.2020.04.009>).

How NGFR was chosen?

We thank the reviewer for the opportunity to clarify the rationale for selecting NGFR as a marker of chronic pancreatitis-like tissue injury.

The relevance of NGFR in this context was, to our knowledge, first reported by Friess *et al.* (Nerve growth factor and its high-affinity receptor in chronic pancreatitis, *Ann Surg.* 1999;230:615–624), who demonstrated its increased expression in chronic pancreatitis, thereby linking NGFR to pancreatic injury. Building on this foundation, NGFR has been part of the routine pathological assessment at our Institution for decades, and we have consistently observed NGFR upregulation in pancreatic lobules under conditions of chronic pancreatitis in our clinical immunohistochemical analyses. We have also observed a consistent reduction of NGFR expression in regions of cancer invasion, suggesting that its expression is linked to a non-cancer-associated stroma in the context of PDAC. Specifically, in our previous work on liver metastases (Moro *et al.*, *Nature Communications*, 2023), we have identified a comparable expression pattern: NGFR-positive stromal cells were prominent in normal portal zones and benign fibrotic conditions (e.g., focal nodular hyperplasia, cholangitis, and cirrhosis, all benign lesions), as well as in the peritumoral stromal rim; in contrast, NGFR expression declined markedly toward the tumor center, where the stroma becomes desmoplastic and “cancer-associated”.

Taken together, these previous studies and routine diagnostics motivated the use of NGFR as a marker for delineating chronic pancreatitis-like tissue injury in this study, which we analyzed further in the new sc RNA seq data. An interesting topic for future studies is the functional role of NGFR⁺ stromal cell populations in benign and malignant disease.

L197 The model accuracy should be reported.

We agree that model accuracy should generally be reported, and we expand on this point further in our response to reviewer #2 below.

It is important to note that the ML approach was not employed for unbiased discovery—where model accuracy would be a primary determinant of analytical rigor—but rather as a practical tool to facilitate the segmentation and annotation of thousands of tissue regions. Manual annotation at this scale would have been prohibitively time-consuming. In our study, to ensure reliability, all ML-generated annotations were subsequently curated by several expert reviewers, with overt errors corrected. This hybrid workflow—combining automated annotations with manual validation—enabled both efficiency and high-quality spatial annotation.

We have clarified our use of the ML-based annotation (e.g. in the abstract, L .42, and in Ll. 191ff). We report model accuracy as determined by the standard metrics of the Aiforia software in Supplementary Table 4. Given our hybrid approach and manual curation of all ROIs, and in order not to confuse the reader, we suggest not including accuracy descriptions in the main text, as they have limited value for judging our data. We have included, as a separate file, **Reviewer Figure 1**, showing all ROIs and all ML-based, pathologist-curated annotations at high resolution.

Fig 3e Please describe how the degree of lobular atrophy was defined.

The degree of atrophy was assessed by a pancreatic pathologist (CFM) and the main author, SS, based on predefined criteria as now described clearer in the Methods section (Ll. 794ff). In response to the reviewer’s comment, we have expanded on the description of the different degrees in the Methods and included the relevant reference (Esposito *et al.*, Guidelines on the

histopathology of chronic pancreatitis, *Pancreatology*, Volume 20, Issue 4, June 2020, Pages 586-593, <https://doi.org/10.1016/j.pan.2020.04.009>).

Fig 4a Did the author divide the distance from 0 to 500um into bins and summarize cell contacts within each distance bin? The legend mentioned there were 5 tumors and 15 ROIs. How the replicates were used, Was the mean or median of tumors (or ROIs) taken?

We appreciate the reviewer's request to clarify the analysis behind Figure 4a. The distance of all tumor cells within pancreatic lobules was measured to the invasion front and rounded to the closest 20µm distance bin. Based on this information, we computed the fraction of the contacting tissues with tumor cells in each distance bin, that is, within each 20µm bin, the fraction of the contacts will sum up to 100%. The figure shows information of all ROIs pooled; for the sake of transparency, in the revised manuscript version, we added Supplementary Figure 13 to show the information for each ROI separately.

Fig 5b Could the two major clones in mouse #3 enriched in tumor-in-stroma and tumor-in-lobule separately? Is there any way to know if any of the clones associated with specific tumor phenotype.

We thank the reviewer for this thoughtful question. As Reviewer #2 raised a closely related point and provided more extensive comments regarding our interpretation of clonality in the context of lobular invasion, we kindly refer to our detailed response to Reviewer #2 below to avoid redundancy, and to our answer to next following point from Reviewer #1.

L317 What evidence would support clonal selection?

We thank the reviewer for this question. In the context of clonal selection, one would expect to observe consistent differences in tumor cell populations that reflect underlying genetic divergence. In the type of data presented in our study, indications of such clonal selection could include compartment-specific expression of protein markers that serve as surrogates for genetic alterations—such as p53 overexpression or SMAD4 loss—between tumor cells in stromal versus lobular regions. Notably, we did not observe any such compartment-restricted differences in protein expression in any of the 31 analyzed human PDAC cases or in the murine models included in this study. The absence of such indicative molecular differences in the human tumors supports our interpretation that microenvironmental cues—rather than clonal selection—are the primary drivers of phenotypic divergence between tumor cells in lobular and stromal tissue compartments.

In the murine orthotopic tumor injection experiments, where we know that one or a few clones make up the whole tumor bulk (Figure 6), it is highly unlikely that one clone of proliferating cells exclusively makes contact with acinar cells at various spatial locations, while the other clone is consistently several cell layers away from the acinar compartment. This would require “acinar-adjacent” clones to give rise to progeny exclusive left and right from the cells in contact with the acinar compartment. Furthermore, upon tumor growth and progression into the lobules, all clonal progeny of cells that had been in contact with acinar cells would have to be eliminated to be able to explain the differences in marker expression (if clonal selection were the only or major driver of subtype heterogeneity), given the way that our marker analyses were conducted.

While comprehensive genomic analyses—such as targeted next-generation sequencing or whole-genome sequencing of microdissected tumor fractions—would be required to approach this question in greater detail, given the confined spaces in which tumor cells reside within the lobules, these would require single-cell level resolution in situ, which is beyond the scope of this study.

Of note, our interpretation of the data does not preclude genetic or tumor cell-intrinsic differences from contributing to subtype heterogeneity, and this model allows for clonal preference to colonize different tissue compartments.

L734 The description of how quantification of tumor cell contacts was not clear. “spatial distance between cell detections and ML-based annotations, together with annotations of invasion front, was computed in QuPath”. Could the author explain how cell detections, ML-based annotation and invasion front annotation were used to calculate distance?

We thank the reviewer for this helpful comment and understand that the complexity of the workflow, combining multiple software platforms and annotation types. We are happy to clarify and have revised the manuscript accordingly (Ll. 584ff) to improve the description of our image analysis approach.

To annotate tissue compartments in human PDAC sections immunolabeled for CD146 and NGFR, we used *Aiforia* (<https://www.aiforia.com/>), a commercially available machine-learning platform. This allowed us to efficiently generate a high number of annotations for the different cell types (e.g., tumor, ADM, acinar cells) that would have been prohibitively time-consuming to produce manually. These ML-generated annotations were then imported into *QuPath* (<https://qupath.github.io/>), an open-source image analysis software offering advanced capabilities for cell detection and spatial quantification. In QuPath, we performed cell segmentation based on nuclear stain to detect individual tumor cells. We also manually annotated the tumor–exocrine invasion front. All annotations—including ML-derived cell types, the manually defined invasion front, and QuPath-based tumor cell detections—were overlaid onto the same whole-slide images. This allowed us to compute the spatial distance from the centroid of each tumor cell to the nearest ML-derived annotation for every cell type and to the invasion front using QuPath’s “Distance to Annotations 2D” tool. We hope this clarifies the methodology, and we are happy to provide additional details if needed. The workflow has been summarized in the Supplementary Figure 13a.

Reviewer #2 (Remarks to the Author):

In this article, the authors investigated possible cellular interactions in the lobular compartment surrounding PDAC. Using cellular classification through machine learning (ML) on whole slide images of human PDAC and some mouse models, the authors describe specific changes at the interface between PDAC and lobules (Lobular changes: ADM, and NGFR+ fibroblasts; PDAC changes: shift towards classical phenotype) and compare these to features of tumour cells in desmoplastic stroma. Based on this, the authors concluded the existence of “an injury-induced tissue niche” in “nascent PDAC invasion”.

The article is interesting and introduces some novel concepts and techniques. However, there are significant weaknesses (as detailed below) that diminish enthusiasm for the paper in its present form. We also have some concerns regarding overreach of conclusions.

We thank the reviewer for their thoughtful comments and critical evaluation of our manuscript. We appreciate the recognition of the novelty of the concepts presented and the concerns raised regarding the interpretation of our findings.

We acknowledge that the original version of the manuscript did not clearly distinguish between spatial and temporal aspects of tumor progression, specifically, between invasion into adjacent tissue (tumor progression across a spatial dimension) and chronological processes such as malignant transformation *in situ* (e.g., in PanIN lesions). In response, we have revised the manuscript to more clearly articulate these distinctions and to avoid conflating spatial relationships, which are the focus of this study, with temporal progression. Examples of these clarifications can be found in the revised text at Ll. 68ff (manuscript version without tracked changes).

As the reviewer rightly notes, spatial and temporal relationships do not necessarily coincide. However, we suggest that spatial patterns, especially when aligned with data from experimental models, can offer meaningful insights into temporal dynamics. For instance, both malignant transformation “in situ” (i.e. the accumulation of mutations without significant proliferation and cell migration) and tumor invasion/migration into adjacent normal tissue involve non-malignant cells encountering oncogenically transformed epithelium. We believe an effort to use parallels between both concepts is valuable for understanding how tumors modulate and remodel their surrounding microenvironment.

We have revised the entire manuscript, focusing on clarifying these conceptual frameworks and ensuring that our conclusions are well-supported and appropriately framed regarding both. Specific comments are addressed in the detailed responses below.

Our specific major comments, in order of importance, are:
1. Assumption that spatial relationships imply temporal relationships lacks convincing experimental support: One of the conclusions of this paper is that the lobular invasion and the features of the cancer/pancreatic lobules in these areas reflect changes in “nascent PDAC invasion”. In other words, the authors are implying these changes reflect the earliest changes in PDAC/pancreas interaction. This conclusion is difficult to justify given the evidence presented in this article represents only a snapshot in time. How can the authors be sure that the tumour cells identified in lobular regions with ADMs are not due to oncogenic changes within existing ADMs in that area?

We thank the reviewer for highlighting the need to better distinguish between spatial and temporal concepts in our manuscript. We have revised the text and manuscript structure to clarify this, including a restructuring of the introduction.

Our interpretation that inflamed lobular regions reflect early stages of PDAC invasion is based on both clinical and experimental observations. Lobular atrophy – **progressing** from acute inflammation to the complete loss of lobular structures – is a well-established phenomenon in non-malignant conditions such as pancreatitis (Esposito et al., Guidelines on the histopathology of chronic pancreatitis, Pancreatology, Volume 20, Issue 4, June 2020, Pages 586-593, <https://doi.org/10.1016/j.pan.2020.04.009>). Given this framework, we reason that acutely inflamed lobules with preserved acinar cells, which we identify and annotate in this study, represent an

earlier stage of tissue injury and atrophy than fully atrophic lobules, where only islets of Langerhans and ductal structures remain (Supplementary Figure 2).

In our human PDAC samples, we performed multiplex immunohistochemistry for p53 and SMAD4 to distinguish tumor cells from adjacent ADM. As shown in Figure 1c, ADM cells adjacent to tumor cells do not exhibit SMAD4 loss and retain normal cytological features, i.e. they lack criteria that are central to the pathological diagnosis of PDAC, which is based on histology alone. While ADM is recognized as a potential precursor to neoplasia, it is generally believed that the progression from ADM to invasive carcinoma occurs over many years to up to one decade in humans (Yachida *et al.*, Nature 2010; <https://doi.org/10.1038/nature09515>), which makes multifocal de novo tumor formation in several invaded lobules highly unlikely.

To further support this interpretation, in the revised manuscript, we analyzed tumors from the KPCT orthotopic mouse model using tdTomato-labeled tumor cells, allowing unambiguous distinction between tumor cells (tdTomato⁺) and host-derived ADM structures (tdTomato⁻). We confirmed the presence of Sox9⁺/tdTomato⁻ cells with characteristic ADM morphology, in wild-type host tissue, adjacent to sites of tumor invasion (new Supplementary Fig. 18). These results demonstrate that non-tumorous ADM exists in the peritumoral lobules and they strengthen our interpretation of spatially restricted, tumor-induced host tissue remodeling.

Additionally, although primary multifocal PDAC can occur, it is relatively uncommon and typically limited in number (Fuyita *et al.*, Pathogenesis of multiple pancreatic cancers involves multicentric carcinogenesis and intrapancreatic metastasis, Cancer Sci. 2020;111(2):739–748; doi:10.1111/cas.14268). In case of multiple intrapancreatic tumors, distinction between independent tumors and intrapancreatic metastases is regularly made based on immunohistochemical profiles (e.g., p53, SMAD4; Fuyita, 2020). Importantly, in the human PDAC samples in our m-IHC cohort (n = 31), we did not observe differential SMAD4/p53 phenotypes suggestive of multifocal neoplasia within the analyzed regions, arguing against, but not ruling out, multifocal PDAC with different origins.

While the concept that PDAC invasion might induce de novo transformation through ADM is intriguing and may relate to relapse mechanisms after surgical resection, the prolonged latency associated with ADM-driven neoplasia, compared to the aggressive timeline of PDAC progression, makes this an unlikely explanation for the lobular tumor phenotype observed in our study. Given these timelines and results from clonality analyses, it is unlikely that ADM cells immediately adjacent to established PDAC in various different lobules (Figure 1) represent newly formed, independent neoplasms, although they might, over the course of several years and if they persist after PDAC insult, give rise to novel tumor lesions. However, most patients with PDAC will not live long enough, given that 5-year overall survival is around 10% (Siegel *et al.*, Cancer statistics, 2025, <https://doi.org/10.3322/caac.21871>).

Taken together, the available clinical and pathological data and the orthotopic inject model support the interpretation that the patterns of lobular invasion described in this study reflect a distinctive growth pattern at the tumor–parenchyma interface rather than the independent emergence of multiple de novo PDAC lesions within each lobule.

The authors have used the spatial relationships between tissue types as a way to deduce temporal relationships. For firmer conclusions, better experimental data are required either by benchtop or animal studies. For instance, how does PDAC phenotype change with animal models with and without stromal co-injection, or in animal models with various degrees of experimentally induced lobular atrophy?

We understand the reviewer's concern regarding the potential conflation of spatial and temporal concepts, and as outlined above, we have revised the manuscript to more carefully distinguish between these dimensions.

The single-cell RNA-seq analyses performed for the revised manuscript, and detailed in response to Reviewer #1, reveal that injury-related changes, such as the emergence of NGFR⁺ stromal cells and ADM formation, occur early after tumor inoculation in an orthotopic tumor cell injection model (new Figure 5g). The number of NGFR⁺ fibroblasts declines as tumors progress, and while classical tumor phenotypes dominate early after injection, basal/mesenchymal tumor phenotypes expand over time. These analyses, while inherently cross-sectional, originate from chronological experiments from tumor-bearing mice at defined early and late timepoints, thereby providing a temporal approximation of tumor progression within the same oncogenic context. Similarly, in human tumors, NGFR⁺ fibroblasts are more abundant in smaller tumor lesions and less abundant in larger tumors (Figure 5h).

To further address the reviewer's important point regarding the chronological development of tumor phenotypes, we analyzed early neoplastic lesions (PanIN) in KC mice (*Kras*-only) versus tumors in KPC (*Kras*, *Trp53* mutated) mice. We found the basal phenotype to be enriched in KPC tumors compared to KC PanINs (Fig. 6a–c). Moreover, in KC mice, HMGA2—a basal marker—was enriched in high-grade (later-stage) PanINs surrounded by desmoplastic stroma, as opposed to low-grade (early-stage) PanINs within slender stroma (new Supplementary Figure 17c&d). This pattern suggests parallels between temporal progression (low- to high-grade PanIN) and spatial transitions observed in our study (lobular to stromal compartments). However, as the reviewer rightly noted, these concepts should not be conflated or generalized, which we have considered in the revised manuscript.

Regarding the reviewer's suggestion to use co-injection of tumor cells with stromal cells, we refer to a previous study performed by some of us (Strell, C., Norberg, K., Mezheyski, A. *et al.* Stroma-regulated HMGA2 is an independent prognostic marker in PDAC and AAC. *Br J Cancer* **117**, 65–77 (2017). <https://doi.org/10.1038/bjc.2017.140>) that used co-injection of human PDAC cells, Panc-1, cancer cells with pancreatic stellate cells. Upon co-injection, the authors found increased tumour growth, associated with increased HMGA2 expression, further supporting our interpretation of the data.

As our data show that PDAC invasion consistently induces lobular inflammation and atrophy, we believe that additional experimental induction of inflammation (e.g., by cerulein injections) would add limited value for understanding the specific phenotypes we describe, which we invariably observe upon tumor invasion.

However, to further study the consequences of tumor-acinar/ADM interactions, we have performed new in vitro co-culture experiments of primary pancreatic acinar cells with PDAC cells. These experiments revealed a phenotypic shift of tumor cells toward a classical-like state when in close proximity to acinar/ADM cells (Fig. 6j–k), providing further evidence for niche-driven modulation of tumor cell phenotype, at least partly driven by tumor-acinar/ADM proximity.

2. Assumption that cellular proximity implies interaction lacks adequate supportive evidence: There is also a question of whether it is always true that proximity implies “interaction”. While logical, this is not guaranteed. For instance, one can argue that the change in PDAC phenotype in the lobular compartment may be due as much to the loss of signals from stroma as it is from close interaction with lobular cells or ADMs. That is, the interaction may be with the cells far away from the leading edge of invasion rather than with cells directly being invaded. For this reason, we believe that this assumption should be backed up by experimental data which support the existence of such interactions or other collateral evidence of interactions (bioinformatics approach).

We fully agree with the reviewer that proximity does not necessarily equate to direct cellular interaction, and that tumor cell phenotypes are likely shaped by a complex array of microenvironmental influences beyond immediate juxtacrine signaling with adjacent cells. A range of factors, including paracrine signaling gradients, changes in extracellular matrix composition and stiffness, hypoxia, and systemic influences such as blood-derived factors, may all contribute to shaping tumor cell behavior within a given tissue context. We thank the reviewer for highlighting the need to emphasize this complexity more clearly. In response, we have revised the manuscript to explicitly acknowledge the multifactorial nature of tumor phenotype regulation (e.g. LI. 171ff, LI 494 ff).

At the same time, to further explore the potential contribution of local **epithelial** interactions, we conducted new in vitro co-culture experiments. These studies demonstrate that proximity to acinar cells can induce a modest but measurable shift toward a classical-like tumor cell phenotype (Fig. 6j–k; Supplementary Fig. 21). These results support the idea that local epithelial interactions may contribute to the observed phenotypic heterogeneity, but do not fully explain it, a point that we now discuss more extensively in the revised manuscript.

3. Validation of ML classification requires strengthening: Given the centrality of the accuracy of the ML classification system to the validity of the study, the authors should provide further data to reassure readers of the accuracy of the ML classification system. This should include both global measures of accuracy, as well as specific measures for specific classes of cells. For instance, Supplementary Figure 11b does not provide any details of the accuracy of any given class of cells, nor a quantified validation of the ML model. This Figure only indicates to the reader that the proportion of fibrosis is much the same between ML and manual annotations. This is because the fibrosis is the major tissue type. But what about other tissue classes (in particular, ADMs)? Also, concordance of overall proportions does not imply accuracy of classification at the cell level, which is more important.

We thank the reviewer for this important comment.

We would like to clarify that our approach does not represent a “traditional” ML pipeline intended for fully independent, unbiased classification based on external ground truth. Rather, the primary goal of using ML in this study was to efficiently generate a high number of initial annotations, which could then be manually reviewed and refined. This strategy allowed us to reduce the otherwise prohibitive manual workload while maintaining high annotation accuracy.

Specifically, we used *Aiforia* to develop a model trained on a limited number of manually annotated regions, following which the ML-generated annotations were imported into *QuPath* for further manual curation and analyses (please see also our response to reviewer #1 to a related question). All ML-generated annotations were systematically curated and refined by two consultant pathologists (CFM and BB) until consensus was achieved. Thus, the final dataset reflects pathologist-validated annotations rather than relying on raw model predictions.

Regarding quality metrics, *Aiforia* provides measures such as sensitivity, precision, F1 score, and area error, based on comparisons between training annotations and model predictions within the same regions. These standard metrics, now included in Supplementary Table 4, offer an estimate of the model's initial performance.

We agree with the reviewer that global concordance of tissue proportions does not substitute for cell-level accuracy. To benchmark model performance at a finer resolution, we compared ML-generated annotations with manual cell-level proximity assessments. In the revised manuscript, we have moved the explanation from the Methods to the main text (L1.) to clarify the workflow: ML-generated annotations served as a first-pass tool, which were subsequently manually curated across all analyzed ROIs. This hybrid process allowed us to combine the efficiency of ML with the precision of expert pathological review.

Importantly, because all annotations used for analysis were manually validated on the same images on which the model was trained, the concept of "ground truth" based on independent validation sets is not directly applicable. Instead, the ML model served as an assistive tool to accelerate annotation, without replacing expert supervision.

We have now explained this approach more clearly in the revised main text (L1. 191 ff) to avoid any misunderstanding regarding the role and validation of the ML-based annotations.

o Along the same line, ADM classification accuracy needs clarification. Firstly, the authors have not presented data related to the validation of ML classification of ADM against ground truth. Second, the accuracy of ground truth is called into question – The study only used 2 markers (KRT19 and AMY) to identify ADMs. We note that a more expanded combination of markers (ADM and others) may have been performed (by multiplex ISH), but it seemed like this was not used for ML training nor for validation. Also, there are no data (apart from representative image in Fig 3f) to show that the identified ADMs using limited markers corresponded to identified ADMs using expanded markers (or even negative for non-ADM markers). It would seem appropriate that the most accurate method of ADM identification (probably combination of morphology + multiplex markers) should be used as “ground truth” for ML validation.

We thank the reviewer for this thoughtful comment and the opportunity to clarify our approach to ADM classification and ML training.

We acknowledge that ADM is a biologically heterogeneous and morphologically complex process, and that no universally accepted molecular marker combination currently exists to unequivocally define ADM, particularly in human pancreatic tissue, where material is limited. While recent studies using in vitro systems (e.g., Jiang *et al.*, Gastro Hep Adv. 2023;2(4):532–543) have begun to propose marker combinations, ADM remains a histomorphological diagnosis: In tissue sections,

it is conventionally defined by the presence of duct-like epithelial structures with reduced zymogen granules emerging within the acinar compartment (Li, Ductal metaplasia in pancreas, *Biochimica et Biophysica Acta, BBA*, - Reviews on Cancer, Volume 1877, Issue 2, March 2022, 188698, <https://doi.org/10.1016/j.bbcan.2022.188698>). In our study, KRT19 upregulation and Amylase (AMY) loss were used as molecular surrogates to approximate the transition from acinar to ductal-like states. AMY⁺/KRT19⁺ double-positive cells, as observed, represent an intermediate or transitional state. We acknowledge, however, that these markers capture only a portion of the biological continuum, and that changes in morphology, marker expression, and differentiation state are unlikely to occur in parallel or in strictly discrete stages.

Regarding the ML classifier, we would like to emphasize that it was trained and curated based on histomorphological features, not on molecular markers. Our ML model was developed to identify nine tissue and cell classes defined by morphological criteria, following traditional pathology-based assessment. After initial training, all ML-generated annotations were manually reviewed and refined by two consultant pathologists to ensure fidelity to morphological definitions, including for ADM regions, as further outlined in our response above. We have included all annotated ROIs in **Reviewer Figure 1** (uploaded as a separate file).

We appreciate the reviewer's suggestion to correlate molecular ADM markers to ML annotations. While direct cell-level overlap was not possible due to the distance between consecutive tissue sections, we observed good overall concordance between regions identified as ADM-rich by our hybrid approach (combining ML-based annotations and manual curation as outlined above), and the ADM marker, MUC6 (**Reviewer Figure 2**; ref. Remmers et al., Aberrant expression of mucin core proteins and o-linked glycans associated with progression of pancreatic cancer. *Clin Cancer Res.* 2013 Apr 15;19(8):1981-93. doi: 10.1158/1078-0432.CCR-12-2662). However, we also observe MUC6 expression in intralobular ductuli in non-inflamed, non-ADM lobules (**Reviewer Figure 2**). These cells, i.e. non-ADM cells, are positive for ADM markers, but, correctly so, are not recognized as ADM by our supervised ML-approach that factors in morphology.

Finally, while we recognize the value of further refining ADM classification using multiplex markers combined with morphology, such an approach was beyond the scope of the current study. We believe that our pragmatic strategy—combining expert morphological annotation, supported by limited molecular surrogates—provides a robust basis for the analysis presented here.

4. Mouse model data can be more convincing: The mouse model did not convincingly demonstrate that the changes in the PDAC cells in the lobules is due to environmental cues rather than being clonal. While the tumour diversity (according to the barcoding prior to injection) suggests the dominance of limited clones, there are no data to support or refute that the changes are subclonal (ie, as a subclone of the dominant barcoded clone). The use of only one marker (HMGA2) to indicate PDAC phenotype is also suboptimal.

We thank the reviewer for raising this important point regarding the distinction between clonal and microenvironmental influences on PDAC phenotypes. We have revised the manuscript to include a broader and more nuanced discussion of the potential interplay between genetic and environmental factors as indicated above.

Our barcoding data demonstrate that orthotopic tumors were predominantly oligo- or monoclonal, with one or a few dominant clones comprising the tumor mass (Figure 6d&e). When comparing tumor cell phenotypes in adjacent regions, specifically, tumor cells within lobules versus those further away from lobular structures, we observed significant phenotypic differences in close spatial proximity (Figure 6f-i). For these differences to arise solely through clonal selection, one would have to assume that distinct clones not only dominate but also strictly segregate anatomically, such that one clone exclusively colonizes lobular regions, and another exclusively colonizes stromal regions. Such strict and complete spatial segregation at the interface versus some cell layers away from the interface without intermixing would be highly improbable, particularly given the continuity of tumor cells observed across the tumor mass and the data arguing for oligoclonality with very low clone numbers from the barcoding experiments.

Recent work on PDAC clonality (e.g., Ho *et al.*, Clonal dominance defines metastatic dissemination in pancreatic cancer. *Science Advances*, 2024, doi: 10.1126/sciadv.add93) has not demonstrated strict spatial segregation of subclones at the PDAC invasion front. Instead, clonal mixing appears to be common, even at the tumor–stroma interface. This further argues against the likelihood of strict clonal partitioning driving the observed spatial phenotypic differences in our model.

Although we primarily used HMGA2 as a basal marker in murine tissue for spatial quantification, we complemented this analysis by including GAL4 as a classical marker. This allowed us to categorize individual tumor cell ROIs as “basal,” “hybrid,” or “classical” based on their basal and classical marker expression, revealing a significant enrichment of classical tumor cells in lobular ROIs (Figure 6g–i; Supplementary Figures 17&19). In addition to GAL4, we also evaluated GATA6, another classical marker, and vimentin, a mesenchymal tumor cell marker representing the most aggressive phenotype along the KPC tumor spectrum (Pitter *et al.*, Systematic Comparison of Pancreatic Ductal Adenocarcinoma Models Identifies a Conserved Highly Plastic Basal Cell State. *Cancer Res.* 2022, 82 (19): 3549–3560. <https://doi.org/10.1158/0008-5472.CAN-22-1742>). However, both markers proved unsuitable for quantitative spatial analysis in FFPE sections: GATA6 showed weak and widespread staining across acinar and most tumor cells, while vimentin exhibited strong stromal staining that precluded accurate tumor cell scoring in both cases.

Together, these findings support a model in which the lobular microenvironment exerts a significant influence on tumor cell phenotypes. While we cannot exclude a contribution from genetic factors, the weight of the experimental and spatial evidence argues against clonal selection being the primary or sole driver of the observed phenotypic divergence.

5. Translational significance is unclear: Finally, the translational implication of these findings is not entirely clear. How do the authors foresee the information from their study being applied to the clinical setting such that it significantly changes patient outcomes? This can be further explored in the discussion.

We thank the reviewer for this important point.

Our central finding is the identification and in-depth characterization of PDAC colonization within pancreatic lobules remodeled by injury and chronic inflammation. Lobular invasion is not

routinely considered during pathological diagnosis or in research studies. Thus, we consider the identification of lobular colonization at substantial frequencies as a pivotal observation that establishes a framework for future investigations into the biological mechanisms underpinning differential invasion patterns across distinct anatomical compartments.

Supporting the relevance of the broader pancreatic tissue context for clinical outcomes, recent studies (e.g., Korpela *et al.*, Pancreatic fibrosis, acinar atrophy and chronic inflammation in surgical specimens associated with survival in patients with resectable pancreatic ductal adenocarcinoma, *BMC Cancer* 2022;22(1):23, doi: 10.1186/s12885-021-09080-0) have demonstrated that fibrosis, acinar atrophy, and chronic inflammation in pancreatic resection specimens are associated with poorer disease-specific survival in patients with resectable PDAC. These findings might suggest that not only tumor-intrinsic factors but also the underlying condition of the pancreatic parenchyma, including its inflammatory status, may critically influence disease progression. In this light, our study linking lobular remodeling and stromal inflammation to PDAC invasion highlights the lobular microenvironment as a potential therapeutic target, motivating further research into lobular colonization, in addition to all ongoing studies on the stromal microenvironment.

From a diagnostic pathology perspective, our results could inform refinements to routine clinical workflows. While PDAC diagnosis is generally based on H&E-stained sections, our findings suggest that incorporating markers such as NGFR (stromal) and p53/SMAD4 (epithelial) may improve diagnostic accuracy, particularly in challenging cases requiring differential diagnosis between PDAC and chronic pancreatitis. For instance, the detection of CD146⁺ ADM-derived ductal cells accompanied by NGFR⁺ stroma is strongly indicative of a reactive, benign process rather than carcinoma. Although such immunohistochemical approaches are standard practice at our institution, they are not widely adopted internationally. We hope our study will raise awareness of the diagnostic value of assessing both epithelial and stromal components, potentially aiding in more accurate diagnosis and margin assessment.

A particularly challenging clinical frontier is the early detection of PDAC. We have revised our discussion to more carefully reflect the limitations and possibilities in this regard (L1. 506ff). While apart from PanIn analyses, most of our study does not directly address early carcinogenesis, the identification of lobular inflammation and early stromal remodeling, including the emergence of NGFR⁺ fibroblasts and ADM, may offer features that could eventually be exploited for early detection strategies. Importantly, PanIN lesions, the most common precursor of PDAC, are rarely detectable by imaging or resected preemptively, and therefore human material for studying these early stages is extremely limited (Kiemer, PanIN or IPMN? Redefining Lesion Size in 3 Dimensions. *Am J Surg Pathol.* 2024 Jul 1;48(7):839-845. doi: 10.1097/PAS.0000000000002245). In this context, recognizing non-desmoplastic lobular growth patterns associated with inflammatory remodeling may help identify early-stage tumors before desmoplastic stroma develops. However, in the revised manuscript, we have taken care to avoid overstating the current clinical applicability of these findings and to frame them as hypotheses that warrant further investigation.

6. Compliance with SAGER guidelines not articulated: For the animal studies, only female mice have been used. This needs to be mentioned in the abstract as per the journal's guidelines.

Thank you for noting this, we have amended the text accordingly (Ll. 48).

Reviewer #2 (Remarks on code availability):

I instructed one of my postdocs to access the provided link and can confirm that the data and code have been provided. However to run the code we would need to update our version of the software which is not currently possible because it requires institutional permission.

Reviewer #3 (Remarks to the Author):

In this manuscript Soderqvist and colleagues describe a distinct injury-associated non desmoplastic niche for pancreatic ductal adenocarcinoma cells in pancreatic lobules. They suggest that the tumor invasion front in the lobules is dominated by interactions between tumor and ADM cells and contains injury activated fibroblasts. In this area PDAC cells expressed a more classical phenotype, whereas PDAC cells with a desmoplastic stroma shows more a basal phenotype. This is an interesting finding which could have implications for understanding how PDAC phenotypes occur or interchange; and could have implications for early detection. However, more evidence needs to be included to fully support these points.

We thank the reviewer for the positive evaluation of our manuscript and their valuable comments.

Macrophages, dependent on their subtype, have been shown to regulate ADM and fibrosis. The subtypes of macrophages in lobular/PDAC and desmoplastic/PDAC are as should be determined and this information should be included into the manuscript.

We have added data from multiplex immunofluorescence of macrophage populations (new Supplementary Figure 16). These data do not identify any clear association between tumor location in lobules or desmoplastic stroma. Nevertheless, they expand on an important and emerging aspect of PDAC biology, and we thank the reviewer for the suggestion. We discuss these data briefly at Ll. 321ff. At the same time, we acknowledge the complexity of macrophage phenotypes that cannot be captured sufficiently within the scope of our manuscript. However, given the inflammatory nature of the lobular mironiche, research into macrophage phenotypes and other immune cell types is highly warranted.

Figure 3F should be accompanied by a H&E staining. Why are there 3 types of ADM cells indicated by different marker combinations?

Thank you for this comment. Acinar cells undergoing ADM, i.e. transitioning to a ductal cell state, express different markers at different stages (Jiang *et al.*, Gastro Hep Adv. 2023;2(4):532–543), but the order of this stepwise process and its heterogeneity are incompletely understood. Our previous multiplex-ISH data shed some light on this heterogeneity. However, as highlighted by this relevant comment from the reviewer, the presence of different ADM states in the context of our study was unclear. Therefore, we have decided to omit the multiplex-ISH results from the current study to avoid confusion. While the ISH results provide insights into ADM heterogeneity,

they are not central to any of the claims of the manuscript and, hence, might be better suited for a follow-up study looking at different responses and stages of acinar transformation.

The interpretation of Figure 4E seems a bit far-fetched. Maybe immunofluorescence for both, NGFR and PDGFalpha, should be shown in addition to Figure 4e (instead of showing this in the Supplemental Figure 15). In Supplemental Figure 15b it is unclear if both markers overlap in their expression. If the NGFR+/PDGFRalpha+ injury-induced stromal cells define a distinct cell population of CAFs (independent of previously described CAF populations) as suggested, they should be isolated and character

We agree that the NGFR⁺ cells in the lobules warranted further characterization. In accordance with this and the suggestion from reviewer #1, we have analyzed NGFR⁺ stromal cells derived from sc RNA-seq datasets from mouse and human PDAC, as well as NGFR⁺ stromal cells in murine pancreatitis. The results from the three independent analyses support the conclusion that NGFR expression overlaps to a significant degree with PDGFRa expression, as described in detail in the response to reviewer #1 above and shown in the new Figures 4c-e, Figures 5e-i, Supplementary Figure 15).

As suggested by the reviewer, we have included images of NGFR/PDGFRa double-positive cells in the revised version of main Figure 5a (former Figure 4e).

The interpretation (Figure 6) is that basal tumor cells, when invading into the injured lobules switch to a classical phenotype and closely interact with ADM cells. But this 'switching' is not really demonstrated and there is no mechanistic insight in how this can occur. Can this be shown ex vivo?

Following suggestions in this line from all reviewers, we conducted *ex vivo* co-cultures of PDAC cells and acinar/ADM cells. Indeed, we observed a significant, albeit modest effect of close proximity of acinar/ADM cells and tumor cells, such that the tumor cells showed an increase of classical and a decrease of basal-like markers close to acinar cells. These experiments support a phenotypic shift of tumor cells toward a classical-like state when in proximity to acinar/ADM cells (Fig. 6j–k), providing evidence for niche-driven modulation of tumor cell phenotype, at least partly driven by tumor-acinar/ADM proximity. Furthermore, additional analyses of sc RNA-seq data from different time points after orthotopic injection of tumor cells link NGFR⁺ stromal cells, ADM, and the classical tumor phenotype (new Figure 6g).

In this regard, it is important to note that we do not mean to claim that tumor-acinar/ADM interactions are the sole drivers of the tumor cell phenotype; rather, it is likely that other factors, including stromal and inflammatory-cell derived factors, but also physical traits such as differences in stiffness in lobular and desmoplastic regions, contribute to the observed changes, which we discuss in more detail in the revised manuscript (e.g. L1 494ff).

Nevertheless, our results establish grounds for further study of tumor-acinar epithelial cell contacts and could allow a more detailed analysis of their crosstalk in future studies.

It also is not sure which phenotype occurs first, which would be important for early detection (as suggested by the authors). If their findings can be reproduced in KPC (as shown) it should be possible to do a time course of tumor formation in KPC and then determine relative changes between phenotypes or more/less classical/lobular areas.

We thank the reviewer for this relevant suggestion. Tumor development and latency are markedly variable between individual KPC mice with autochthonous tumor development (Spadafora, *Vet al. BMC Cancer* 24, 414 (2024). <https://doi.org/10.1186/s12885-024-12104-0>), which is why comparing differently aged KPC mice would not sufficiently answer the reviewer's valid point, since we would not be able to determine the time point of tumor development in each individual lesion (i.e., we might conflate chronological and temporal relationships). We kindly also refer the reviewer to our answer to reviewer #2 above regarding this distinction.

To address this point, we used two other approaches, as also outlined in response to reviewer #2 above:

1. We analyzed early neoplastic lesions (PanIN) in KC mice (*Kras*-only) versus tumors (PDAC) in KPC (*Kras*, *Trp53* mutated) mice. We found the basal tumor cell phenotype to be enriched in KPC tumors compared to KC PanINs (Fig. 6a–c). Moreover, in KC mice, HMGA2, a basal marker, was enriched in high-grade (later-stage) PanINs surrounded by desmoplastic stroma, as opposed to low-grade (early-stage) PanINs within slender stroma (new Supplementary Figure 17c&d).
2. We analyzed human and murine PDAC single-cell RNA sequencing datasets. The new data are presented in Figure 5e–I and Supplementary Figure 15. Specifically, we analyzed data from orthotopic tumor cell injections, derived from different time points after injection (days 10, 20, and 30). We find NGFR⁺ stromal cell enrichment and concordant enrichment of the classical tumor cell phenotype **early** after tumor cell injection (new Figure 5e). We also observe NGFR⁺ stromal cells to be enriched in smaller human PDAC (new Figure 5h), while no enrichment of classical human PDAC cells became evident, which might be related to general difficulties in sampling human PDAC for sc-seq or reflect inter-patient heterogeneity of human tumors.

Together, these data argue for a concordant progressive loss of classical tumor cells traits and NGFR⁺ stromal cell signatures.

In Figure 1C the SMAD4 IHC is not convincing.

We have provided an alternative stain following the reviewer's suggestion. We would be happy to provide anonymized whole slide images of SMAD4 or any other stains to the reviewers.

Reviewer #4 (Remarks to the Author):

Response to reviewer comments, NCOMMS-24-41478A

Thank you for co-reviewing our manuscript.

Reviewer Figure 1. Machine learning–based annotations. This file contains all annotations generated using Aiforia and curated by pathologists. Each case is labeled with a pseudonymized tumor ID (T#), and the tumor region is classified as either “stromal” or “lobular,” as indicated in each image. Annotated regions are shown on the left, with the corresponding unannotated regions on the right. The images begin on the following page.

Tissue classes

- | | | | | |
|--------------|-------------------|----------------------|--------|---------|
| acinar cells | fibroblasts & ECM | ductal cells | nerves | vessels |
| ADM | immune infiltrate | islets of Langerhans | tumor | |

Tissue classes

acinar cells	fibroblasts & ECM	ductal cells	nerves	vessels
ADM	immune infiltrate	islets of Langerhans	tumor	

Tissue classes

acinar cells	fibroblasts & ECM	ductal cells	nerves	vessels
ADM	immune infiltrate	islets of Langerhans	tumor	

Tissue classes

- | | | | | |
|--------------|-------------------|----------------------|--------|---------|
| acinar cells | fibroblasts & ECM | ductal cells | nerves | vessels |
| ADM | immune infiltrate | islets of Langerhans | tumor | |

Tissue classes

- | | | | | |
|--------------|-------------------|----------------------|--------|---------|
| acinar cells | fibroblasts & ECM | ductal cells | nerves | vessels |
| ADM | immune infiltrate | islets of Langerhans | tumor | |

Tissue classes

acinar cells	fibroblasts & ECM	ductal cells	nerves	vessels
ADM	immune infiltrate	islets of Langerhans	tumor	

Tissue classes

acinar cells	fibroblasts & ECM	ductal cells	nerves	vessels
ADM	immune infiltrate	islets of Langerhans	tumor	

Tissue classes

- | | | | | |
|--|---|--|--|---|
|  acinar cells |  fibroblasts & ECM |  ductal cells |  nerves |  vessels |
|  ADM |  immune infiltrate |  islets of Langerhans |  tumor | |

Tissue classes

acinar cells	fibroblasts & ECM	ductal cells	nerves	vessels
ADM	immune infiltrate	islets of Langerhans	tumor	

Tissue classes

acinar cells	fibroblasts & ECM	ductal cells	nerves	vessels
ADM	immune infiltrate	islets of Langerhans	tumor	

Tissue classes

acinar cells	fibroblasts & ECM	ductal cells	nerves	vessels
ADM	immune infiltrate	islets of Langerhans	tumor	

Tissue classes

 acinar cells	 fibroblasts & ECM	 ductal cells	 nerves	 vessels
 ADM	 immune infiltrate	 islets of Langerhans	 tumor	

Tissue classes

- | | | | | |
|--------------|-------------------|----------------------|--------|---------|
| acinar cells | fibroblasts & ECM | ductal cells | nerves | vessels |
| ADM | immune infiltrate | islets of Langerhans | tumor | |

Tissue classes

acinar cells	fibroblasts & ECM	ductal cells	nerves	vessels
ADM	immune infiltrate	islets of Langerhans	tumor	

Tissue classes

acinar cells	fibroblasts & ECM	ductal cells	nerves	vessels
ADM	immune infiltrate	islets of Langerhans	tumor	

Reviewer figure 2: Overlapping regions of MUC6 and Aiforia-annotated tissue classes. Left panels: Mucin 5 oligomeric mucus/gel forming 5AC (MUC5AC; brown) co-stained with ADM marker MUC6 (red). Right panels: Cluster of differentiation (CD)146 (brown) co-stained with nerve growth factor receptor (NGFR; red). a) Whole-slide images (WSIs) of two consecutive sections from one of the tumors in the machine learning pipeline. b) magnified area of a well-preserved lobule, where MUC6+ cells represent centroacinar cells (black arrowheads). c) Magnified area of a lobule, which was subject to calling the indicated tissue classes by Aiforia, with (top right) and without (bottom right) the annotation overlay. Here, the MUC6+ area overlaps with the CD146+ ADM-dominant area on the consecutive section (blue arrows and markings).

Stockholm, July 13th, 2025

Dear Reviewers:

We thank you once again for the constructive and insightful comments on our manuscript. Please find point-by-point responses to the remaining points below.

Reviewer #1 (Remarks to the Author):

The authors addressed all my comments.

I have a few additional comments on the revised manuscript:

Figure 2e. The colors used for the row annotation track of marker gene are not well distinguishable, making it hard to see the trend. Alternatively, the author can consider split rows by marker. It also helps to see the heterogeneity across tumors.

Thank you for your helpful comments. We have amended the colors in this Figure to improve visibility, including a new color palette for the marker metadata. The whole Figure panel has been enlarged.

L294 The second part of the paragraph is repeated.

We apologize for this oversight, which we have now corrected by removing the repetition.

Some of the supplemental figures have low resolution, the text is not readable, for example supp fig 15d.

Thank you for pointing this out. We assume that this might have been aggravated by further processing of the PDF file after submission, and we have carefully checked for readability in the Supplementary Figures.

L353, why different markers for classical tumor are used here? CDX2 and MUC5AC were used at the beginning instead of GAL4, GATA6.

The use of different markers in mouse tissue was based on two considerations; firstly, our previous experience with GATA6 in murine tissues (first submission) supported its use in murine PDAC. Secondly, analyses of murine sc-seq data identified GAL4 as a classical marker that allowed particularly good differentiation between classical, basal, and mesenchymal phenotypes, as outlined in the paragraph, and a validated antibody was available. In contrast, antibodies against CDX2 and MUC5AC murine epitopes were not established in our hands, and the analyses of public expression data indicated that they are expressed in only a few murine tumor cells (see comment below). Hence, we chose GATA6 and GAL4 for the availability of reliable antibodies and their expression profile in an independent dataset.

L359-362 It is hard to see the trend as stated "GAL4 and HMGA2 were expressed with limited overlap in classical and basal/mesenchymal tumor cells" in Supp Fig 17b. GAL6 and GATA6 are classical markers, and should they overlap with classical signature? Are they part of the Pitter et al classical signature?

Thank you for pointing this out. Yes, *GAL6* and *GATA6* are part of the Pitter et al signature, *GAL4* #374 & *GATA6* #946 in the gene list of Pitter et al.). However, *CDX2* and *MUC5* are expressed at low levels by the murine tumor cells and, therefore, were not pursued by us for the murine tumors. Generally, the data presented in Suppl Fig 17b suggested that differences in classical and basal/mesenchymal marker expression are more homogenous and less distinct than in human tumors; we agree that a more careful wording would be more appropriate. We have rephrased this sentence slightly to more accurately reflect the marker expression: “*Of note, after orthotopic injection, murine sc RNA-seq data revealed GAL4 and HMGA2 to be expressed at relatively distinct poles corresponding to classical and basal/mesenchymal tumor cell clusters, respectively, while GATA6 and VIM were more broadly expressed (Supplementary Fig. 17b), supporting the usefulness of GAL4/HMGA2 specifically to call murine PDAC phenotypes in vivo.*”

Reviewer #2 (Remarks to the Author):

The authors have satisfactorily addressed some of the comments but it would be useful if the following remaining issues relating to comments 1,2 and 3a could be further addressed:

We thank the reviewer for their careful assessment of our manuscript.

Comments 1 and 2

The provided explanation as to why tumour cells in lobular regions are not multifocal de novo tumor seems adequate and the use of tdTomato labelled cancer cells in mouse model is supportive. The comparison between PanIN in KC mice and tumours in KPC mice is useful, but does not necessarily imply the temporal change over time (i.e., lobular compartment being early -> stromal compartment being late; which corresponds to change in phenotype from classical-> basal subtype), especially given PanINs are non-invasive (i.e., comparison between cancer and non-cancer).

We fully agree with this interpretation and have clarified in the text that PanINs are precancerous lesions and that these data are not intended to support chronological inferences. We would also like to note that these observations remain consistent with the main findings of the manuscript, which are supported by both in vivo and in vitro data.

The in vitro data provided 6j-k are interesting, but despite the high statistical significance (this is due to the large number of cells analysed), the effect size does not seem large (from 6k: the violin plots of and the small change in the median marker intensity).

This suggests that the phenotype differences between in-lobule and in-stroma cancer is only partly explained by interaction with ADM. Therefore, the language around the implied temporal relationship/causality needs to be toned down

- classical at early lobular interaction -> basal at late stage, in stroma;
- cancer invasion -> lobular atrophy and ADM; and
- ADM causes cancer to be classical in subtype.

We concur entirely with this interpretation, and we have amended the last paragraph of the results section accordingly. We have revisited the Discussion and slightly amended the section discussing

the strength of the effect: “...interactions within epithelium-rich compartments may contribute to shaping tumor cell behavior and potentially influence therapeutic responses in a tissue context–dependent manner”.

Comment 3a

The provided explanation as to why tumour cells in lobular regions are not multifocal de novo tumor seems adequate and the use of tdTomato labelled cancer cells in mouse model is supportive. The comparison between PanIN in KC mice and tumours in KPC mice is useful, but does not necessarily imply the temporal change over time (i.e., lobular compartment being early -> stromal compartment being late; which corresponds to change in phenotype from classical-> basal subtype), especially given PanINs are non-invasive (i.e., comparison between cancer and non-cancer).

The in vitro data provided 6j-k are interesting, but despite the high statistical significance (this is due to the large number of cells analysed), the effect size does not seem large (from 6k: the violin plots of and the small change in the median marker intensity).

This suggests that the phenotype differences between in-lobule and in-stroma cancer is only partly explained by interaction with ADM. Therefore, the language around the implied temporal relationship/causality should be toned down

- cancer invasion -> lobular atrophy and ADM; and*
- ADM causes cancer to be classical in subtype.*

We thank the reviewer for these comments, which we have addressed in answer to the comment above.

Reviewer #2 (Remarks on code availability):

I instructed one of my postdocs to access the provided link and can confirm that the data and code have been provided. However to run the code we would need to update our version of the software which is not currently possible because it requires institutional permission.

Reviewer #4 (Remarks to the Author):

We thank the reviewers for their valuable feedback on our manuscript.